# Identification of differentially recognized T cell epitopes in the spectrum of tuberculosis infection

Sudhasini Panda [1], Jeffrey Morgan [1], Catherine Cheng [1], Mayuko Saito[2], Robert H. Gilman[3,4], Nelly Ciobanu[5], Valeriu Crudu [5], Donald G. Catanzaro [6], Antonino Catanzaro[7], Timothy Rodwell[7], Judy S. B. Perera[8], Teshan Chathuranga[8], Bandu Gunasena[9], Aruna D. DeSilva[1,8], Bjoern Peters [1,7], Alessandro Sette [1,7,10] & Cecilia S. Lindestam Arlehamn [1,10] ✉

There is still incomplete knowledge of which *Mycobacterium tuberculosis* (*Mtb*) antigens can trigger distinct T cell responses at different stages of infection. Here, a proteome-wide screen of 20,610 *Mtb*-derived peptides in 21 patients mid-treatment for active tuberculosis (ATB) reveals IFNγ-specific T cell responses against 137 unique epitopes. Of these, 16% are recognized by two or more participants and predominantly derived from cell wall and cell processes antigens. There is differential recognition of antigens, including TB vaccine candidate antigens, between ATB participants and interferon-gamma release assay (IGRA + /−) individuals. We developed an ATB-specific peptide pool (ATB116) consisting of epitopes exclusively recognized by ATB participants. This pool can distinguish patients with pulmonary ATB from IGRA + /− individuals from various geographical locations, with a sensitivity of over 60% and a specificity exceeding 80%. This proteome-wide screen of T cell reactivity identified infection stage-specific epitopes and antigens for potential use in diagnostics and measuring *Mtb*-specific immune responses.

Tuberculosis is the ninth leading cause of death worldwide and the leading cause from a single infectious agent, ranking above HIV/AIDS. The World Health Organization (WHO) estimates that approximately one-quarter of the world's population (1.7 billion total) is infected with *Mycobacterium tuberculosis* (*Mtb*). In 2021 *Mtb* was responsible for 1.6 million deaths and ~10 million new infections[1]

Infection with *Mtb* manifests as a spectrum of diseases ranging from asymptomatic subclinical infection (latent infection, LTBI) to active disease (ATB)[2,3]. To date, tuberculin skin test (TST) and interferon-gamma release assays (IGRA), which detect an immunological response against *Mtb*, are diagnostic tests for latent infection[4,5]. Importantly, the classically used tuberculin skin test can be positive in both LTBI and BCG vaccinated participants and thus cannot distinguish them[6,7]. A major advance in the diagnosis of *Mtb* infection was represented by the introduction of IGRA tests that consist of ex vivo analysis of peripheral blood cells for a cytokine (IFNγ) response to peptide pools spanning the ESAT-6 and CFP10 antigens recognized by T cells[8]. As these antigens are absent from *M. bovis* BCG and most nontuberculous mycobacteria (NTMs), such responses can distinguish prior or current *Mtb* infection from BCG vaccination and NTM

[1]Center for Infectious Disease and Vaccine Research, La Jolla Institute for Immunology, La Jolla, CA, USA. [2]Department of Virology, Tohoku University Graduate School of Medicine, Sendai, Japan. [3]Johns Hopkins School of Public Health, Baltimore, MD, USA. [4]Universidad Peruana Cayetano Heredia, Lima, Peru. [5]Phthisiopneumology Institute, Chisinau, Republic of Moldova. [6]Department of Biological Sciences, University of Arkansas, Fayetteville, AR, USA. [7]Department of Medicine, University of California San Diego, La Jolla, CA, USA. [8]Faculty of Medicine, General Sir John Kotelawala Defense University, Ratmalana, Sri Lanka. [9]National Hospital for Respiratory Diseases, Welisara, Sri Lanka. [10]These authors contributed equally: Alessandro Sette, Cecilia S. Lindestam Arlehamn. ✉e-mail: cecilia@lji.org

diseases. However, these tests cannot reliably discriminate between active TB and LTBI[9]. These observations indicate the need for a more extensive search for a panel of antigens that can distinguish LTBI and active disease.

T cells are critical in the host immune response against *Mtb* infection. The responses against *Mtb* infection involve classically restricted CD4 and CD8 αβ T cells[10,11] and non-classically restricted T cells such as NKT (CD1), MAIT (MR1), and γδ T cells[12–14]. Among these, CD4 T cells represent a major component of T cell response against *Mtb* infection[15]. Individuals with low levels of CD4 T cells, such as those who are HIV positive, are more vulnerable to both primary and reactivation of TB[16]. Multiple studies have revealed the recognition of T-cell epitopes from *Mtb* among diverse populations, including individuals with TB disease, LTBI, BCG vaccination, or exposure to *Mtb* and/or NTM[17–20], also reviewed in ref. 21. T cell responses at the antigen and epitope levels are highly complex, particularly in complex organisms. They can involve multiple antigens and hundreds of epitopes[22–25], with broader patterns of immunodominance observed in humans compared to genetically homogeneous murine model[26–28]. While the mechanisms underlying T cell immunodominance and breadth have been extensively studied in murine models and, to some extent, in humans[22,29–32], a quantitative assessment of the complexity of responses during natural infections at the population level is currently lacking. Most immuno-profiling studies have focused on individual antigens or a limited set of epitopes, assuming they represent the entire pathogen-specific response[33,34]. It remains uncertain to what extent underestimating the true complexity of these responses could affect the outcomes of immuno-profiling studies. Therefore, a comprehensive understanding of *Mtb* epitopes across TB disease states is crucial. *Mtb* expresses different proteins during various stages of infection, resulting in stage-specific immune responses. Thus, identifying novel CD4 T cell epitopes at different stages of TB could aid in discovering new diagnostic and vaccine candidates against TB. Previously, our team conducted a proteome-wide screen to detect HLA class II restricted epitopes and antigens recognized in healthy IGRA+ participants without any signs of active TB[22]. This screen included 4000 ORFs of the *Mtb* genome and approximately 20,000 predicted HLA class II epitopes. T cell reactivity against these epitopes was measured using ELISPOT to detect IFNγ response ex vivo. It is equally important to screen these epitopes in ATB disease. Identifying the ATB-specific antigens will be important as boosting immune responses against antigens expressed during the active phase of infection might help translate into reduced incidence or reactivation of infection.

This study explored the possibility that distinct *Mtb* disease states (i.e., healthy IGRA+ participants and patients with active pulmonary TB) respond to differential *Mtb*-derived epitopes and/or elicit different *Mtb*-specific T cell responses. We hypothesized that this differential reactivity could be attributed to variations in response magnitude, specificity (i.e., which antigens are recognized), breadth (i.e., how diverse the response is), and the phenotype of the specific T cells. By conducting a comprehensive proteome-wide screening for *Mtb*-derived T cell epitopes, we identified and characterized epitopes specifically recognized by participants with active pulmonary TB, suggesting their potential utility as diagnostic markers and vaccination candidates.

## Results

### Broad and heterogeneous T cell responses against Mtb-derived HLA class II predicted epitopes in participants with ATB

Detailed knowledge regarding the antigens recognized in TB disease and associated phenotypes is relevant for understanding TB immunopathology and for vaccine and diagnostic applications. alike. We previously used a proteome-wide peptide library of 20,610 *Mtb*-derived 15-mer peptides predicted promiscuous HLA class II binders to define HLA class II restricted epitopes and associated antigens in healthy IGRA+ participants (i.e., LTBI) from San Diego, US[22]. Here, we used the same comprehensive library to determine the repertoire of T cell antigens and epitopes recognized in ATB.

The proteome-wide peptide library was arranged into 1036 peptide pools of 20 peptides each. T cell reactivity against the 1036 peptide pools was measured by ex vivo production of IFNγ and IL-17 using Fluorospot assays and PBMCs from 21 participants mid-treatment for their ATB infection (3–4 months post diagnosis) recruited from the Universidad Peruana Cayetano Heredia site (Peru). This mid-treatment timepoint was selected based on the participants being eligible for a leukapheresis donation, thus allowing the entire peptide library to be tested in each participant. IFNγ was selected based on the important role of this cytokine in immune responses against Mtb[35–38], similarly, IL-17 plays a role in TB pathogenesis[39].

No IL-17 was detected in response to the peptide pool stimulations. Therefore, we selected peptide pools based on IFNγ response for deconvolution to identify individual epitopes. A total of 78 unique peptide pools were selected for deconvolution, corresponding to the ten peptide pools with the highest response magnitude per participant and/or that were recognized by at least two different participants. A total of 137 individual epitopes were identified (Supplementary Data 1). Each participant recognized an average of 8 unique epitopes (range 1–27, median 7), underlining the breadth of responses to *Mtb* (Fig. 1A). Among the 137 individual epitopes, 22 (16%) were recognized by multiple participants (Fig. 1B). When epitopes were ranked based on magnitude, the top 55 epitopes accounted for 80% of the total response (Fig. 1C). In conclusion, the breadth of responses detected in ATB is broad, although narrower ($p = 0.002$, two-tailed unpaired Student's t-test) compared to the previous results in healthy IGRA+ participants who recognized, on average, 24 epitopes[22]. These identified epitopes from the proteome-wide screen in participants with ATB have been submitted to IEDB (Immune Epitope Database) and can be found under the submission ID: 1000914.

### Distinct immunodominant antigens predominantly from cell wall and cell processes in ATB participants

The epitopes were mapped to individual *Mtb* ORFs using H37Rv as a reference genome. A total of 97 ORFs were recognized, with each participant recognizing, on average, 6 ORFs (median 5, Fig. 2A). As expected, the well-known antigens Rv0288 (TB10.4), Rv3875 (ESAT-6), Rv3874 (CFP10), and Rv3615c (included in the "ESAT-6 free" IGRA test[40] were the most frequently recognized. However, nine novel antigens, which have not been previously described as antigens for *Mtb*, were also identified (Supplementary Data 1).

Using the Mycobrowser tool[41], we determined the protein categories to which the identified antigenic ORFs belonged. As previously observed for IGRA+ participants[42], essentially every protein category was represented (Fig. 2B). However, the ORFs antigenic in ATB were predominantly found in the cell wall and cell processes categories, followed by conserved hypotheticals, PE/PPE, and intermediary metabolism and respiration categories (Fig. 2B). Compared to the H37Rv genome, the antigenic ORFs were overrepresented in the cell wall, cell processes category, and the PE/PPE categories and underrepresented in conserved hypotheticals (Fig. 2C). The overrepresentation of the cell wall and cell processes category appears to be specific for the ATB cohort because these antigen categories were not overrepresented in the previous screen of IGRA+ participants[22].

Reactivity in healthy IGRA+ participants[22] previously identified three "antigenic islands" of clustered antigen genes that comprised secreted and non-secreted *Mtb* proteins involved in type 7 secretion systems. Indeed, all three antigenic islands were also identified in the present screen of ATB participants. (Fig. 2D). In contrast to IGRA+ participants in the previous screen, the breadth of responses to antigens targeting the antigenic islands was narrower in the participants with ATB (Fig. 2E). For example, for island 1, only 3 antigens

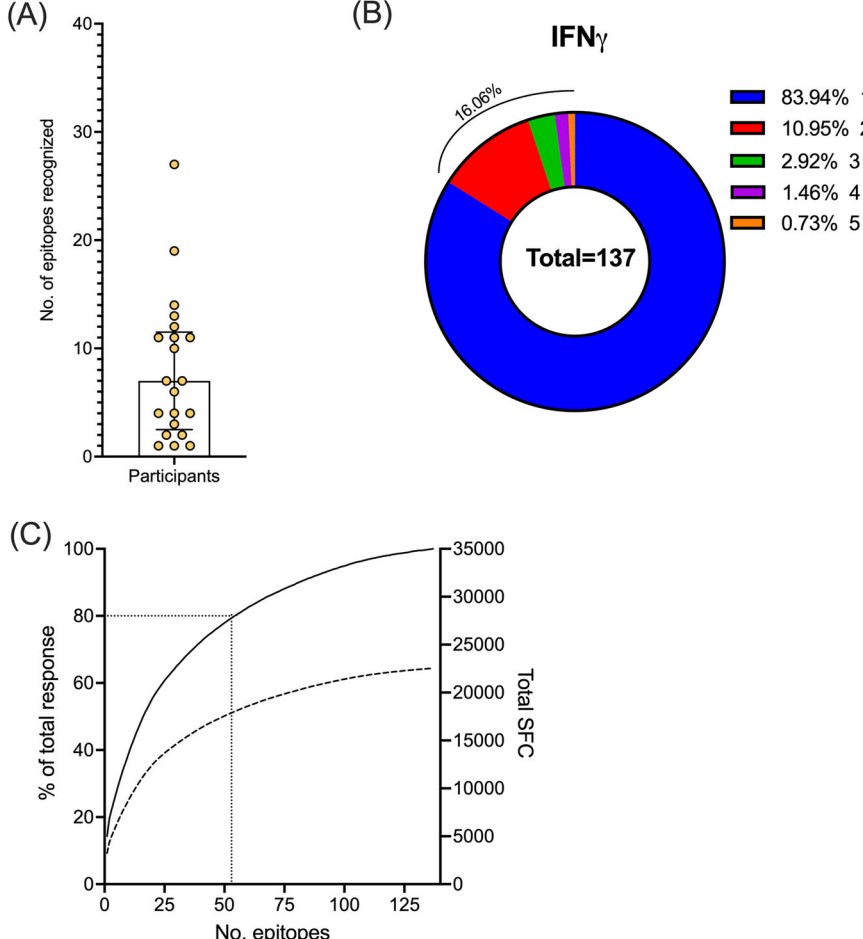

**Fig. 1 | Breadth and dominance of epitopes in mid-treatment ATB participants.**
**A** Number of epitopes recognized by each participant. Each dot is one participant,
$n = 21$; median ± interquartile range is shown. **B** Distribution of recognized epitopes
by the number of participants recognizing each epitope. **C** Epitopes ranked based
on the magnitude of response (solid line - % of total response, dotted line – total
spot forming cells (SFC)). Black dotted lines indicate the top 55 epitopes. Source
data are provided as a Source data file.

represented in the predicted peptide library of 20,610 peptides were
recognized by participants with ATB, compared to 9 recognized by
healthy IGRA+ participants. The same is true for antigens in islands 2
and 3. In conclusion, the data suggest that different patterns of anti-
genic ORFs might be associated with ATB vs. healthy IGRA+
participants.

## Hierarchy in reactivity against TB Vaccine and IGRA antigens

Previous studies showed that the proteome-wide library of predicted
promiscuous HLA class II binders captures about 50% of the total
reactivity[22,43]. To further evaluate T cell responses against TB vaccine
candidate and IGRA antigens (see methods), we tested 517 15-mer
peptides overlapping by 10 amino acids spanning these antigens.
Positive pools were deconvoluted to identify individual T cell epitopes.
Overall, 67% of the ATB participants recognized epitopes from at least
one antigen; on average, these participants recognized 2 different
antigens (range 1–4). This is similar to our previous reports high-
lighting the inter-individual variability of epitope-specific responses[42].

We next compared the magnitude and frequency of the response
for these antigens in the ATB participants with what was previously
observed in healthy IGRA+ participants from South Africa[42]. The most
frequently recognized antigens in ATB and IGRA+ alike were Rv0288
(TB10.4), Rv3875 (ESAT-6), Rv3874 (CFP10), and Rv1196 (PPE18)
(Fig. 3A, B). However, some antigens were differentially recognized in
the two cohorts. Specifically, Rv3875 was more frequently recognized
than Rv3874 in ATB vs. IGRA+ participants. The Rv1813c antigen was

reactive in participants with ATB and not in IGRA + . Finally, the
Rv3619c (EsxV), Rv2660c, Rv0125 (Mtb32a), and Rv2608 (PPE42)
antigens were reactive in IGRA+ and completely unreactive in partici-
pants with ATB (Fig. 3A, B).

The proteome-wide screen detected the two most reactive vac-
cine antigens, Rv0288 and Rv3874. Rv3875 was not detected in the
proteome-wide screen, likely due to its small size, with only 2 peptides
representing it in the proteome-wide library. The results confirm that
the antigens detected in the proteome-wide screen are the most fre-
quently recognized together with Rv3875 and that the other vaccine
antigens account for a small fraction of the response in this ATB
cohort. In conclusion, the screen of vaccine and IGRA antigens, toge-
ther with the proteome-wide screen, identified a total of 174 epitopes,
which were next investigated for functionality and differential reac-
tivity in further experiments.

## Differential patterns of cytokine expression and memory sub-
## sets in CD4 T cell responses against different protein antigen
## categories

The responses associated with the 174 identified epitopes were char-
acterized in more detail in a randomly selected subset of the partici-
pants with ATB (mid-treatment) from Peru by intracellular cytokine
secretion assays to characterize T cell responses specific to antigens
from the different protein categories. One epitope pool (PC85) cor-
responded to epitopes from antigens in the cell wall and cell processes
category, and a separate pool encompassed epitopes from other

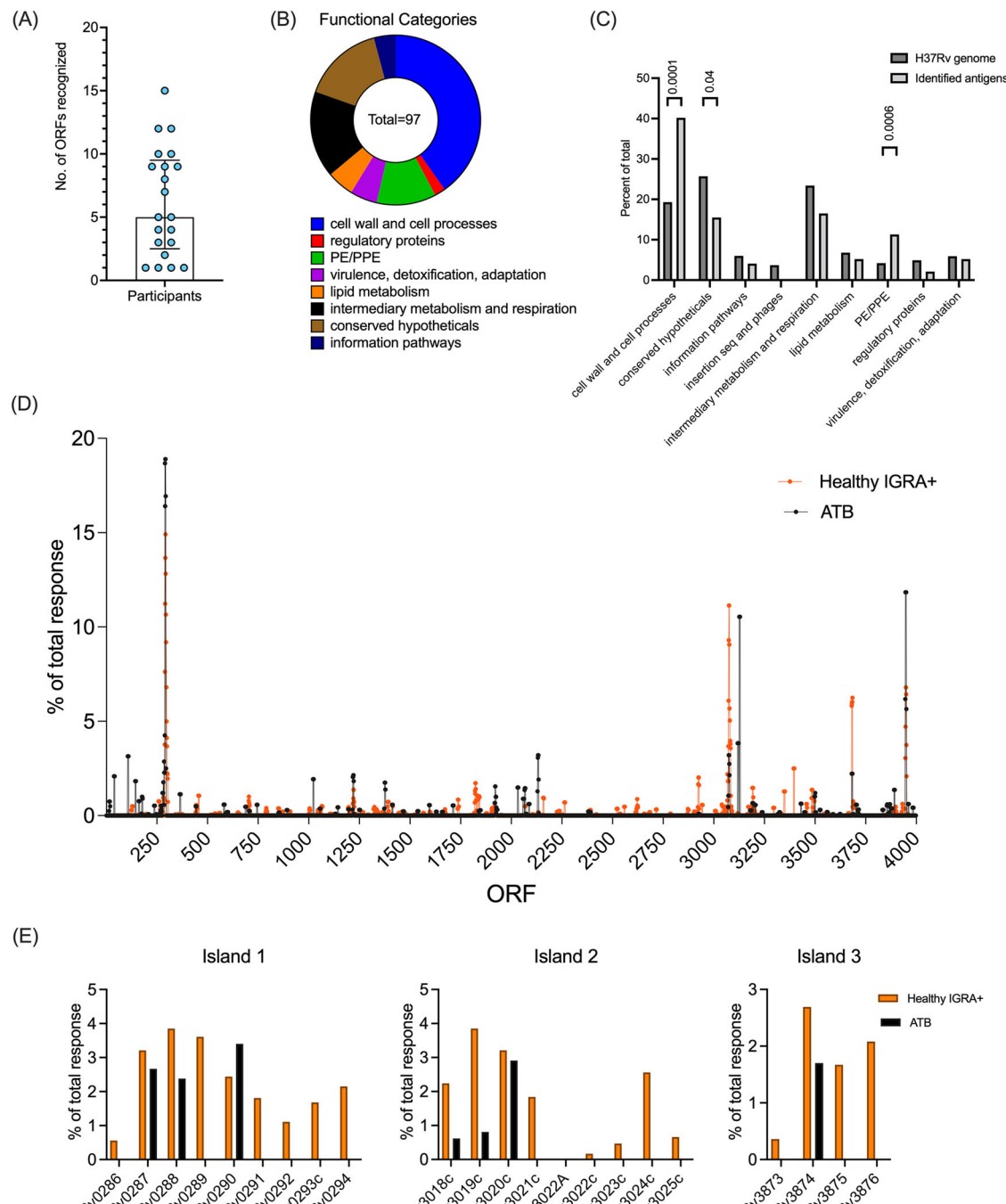

**Fig. 2 | Immunodominant antigens in ATB. A** Number of open reading frames (ORFs) corresponding to recognized epitopes by each participant. Each dot represents one participant, $n = 21$, median ± interquartile range is shown. **B** Distribution of recognized ORFs per protein category. **C** The identified antigens (black bars) were divided into protein categories (Mycobrowser) and compared to the H37Rv genome (grey bars). Chi-square test (two-sided). **D** Antigenic islands identified by a 5-gene window spanning the H37Rv genome. ATB (black) compared to healthy IGRA+ (orange[22],). **E** Proteins within each antigenic island, % of total response per antigen across the proteome-wide screen. ATB (black bars) and healthy IGRA+ (orange bars). Source data are provided as a Source data file.

protein categories (PC71) (Supplementary Data 1). As a comparator, we included the previously described MTB300 pool, based on epitopes recognized in IGRA+ participants[22], which includes peptides from all functional categories.

As expected, based on the peptide library design to bind HLA class II alleles, most of the response was mediated by CD4 T cells (Fig. 4A). Among them, the frequency of TNFα + CD4 cells was relatively higher than IFNγ ($p = 0.0006$ for PC71) or IL-2 ($p = 0.04$ for PC85) (Fig. 4B). In addition, some responses were detected in CD8 + T cells (Fig. 4B), likely due to nested HLA class I binding epitopes within the 15-mer peptides.

Frequencies of cytokine-expressing CD4 T cells in response to both pools were similar to MTB300 (IFNγ, $p = 0.98$ for PC85 and PC71; IL-2, $p = 0.31$ and 0.97 for PC85 and PC71 respectively; TNFα, $p = 0.78$ and 0.51 for PC85 and PC71 respectively) (Fig. 4A, gating strategy shown in Figure S1). The vast majority of cytokine producing CD4 T cells expressed a single cytokine TNFα or IFNγ, followed by TNFα + IL2 + and TNFα + IFNγ + dual cytokine expressed cells with none to very few IFNγ + IL2 + cells and triple cytokine producing cells (Fig. 4C, D). A higher frequency of single TNFα-producing T cells was present compared to other single (IFNγ $p = 0.01$ (PC71), IL-2 $p = 0.01$ (PC85)),

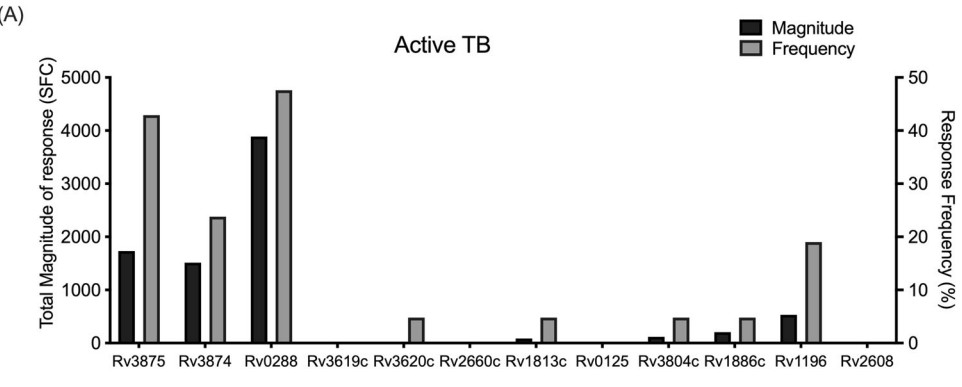

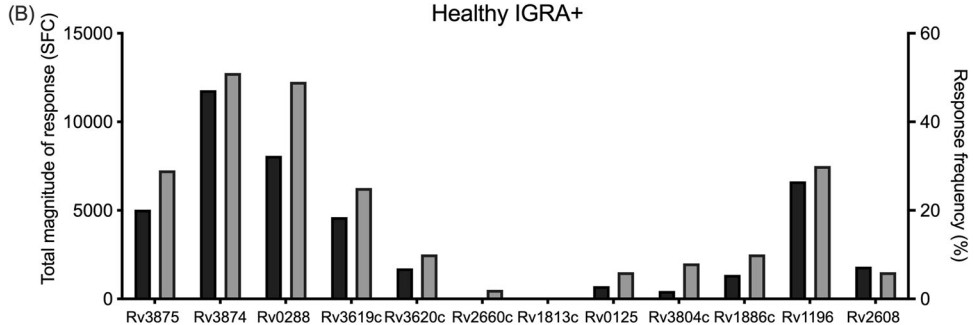

**Fig. 3 | Hierarchy in T cell reactivity against TB vaccine and IGRA antigens.** Magnitude of response, expressed as the total magnitude of response (black bars, left y-axis) or frequency of participants responding (grey bars, right y-axis), amongst the participants. **A** ATB, $n = 21$. **B** Healthy IGRA + , $n = 63$[42], for comparison purposes. Rv number for each antigen are indicated on the x-axis. Source data are provided as a Source data file.

dual (IFNγ + IL-2+ p = 0.0006 (PC71), $p = 0.0002$ (PC85), TNFα + IFNγ + $p = 0.002$ (PC71), $p = 0.01$ (PC85), TNFα + IL2 + p = 0.02 (PC71), $p = 0.008$ (PC85)), or triple ($p = 0.0008$ (PC71), $p = 0.0003$ (PC85) cytokine producing cells (Fig. 4C).

Next, we characterized which T cell memory subset was responsible for the reactivity. Memory subset phenotypes were determined using antibodies to CD45RA and CCR7. The majority of the cytokine producing CD4 T cells were effector memory (CD45RA-CCR7-), and as expected few were naïve (CD45RA + CCR7 + ) cells (Fig. 4E, F). For IFNγ there was no difference between the CD4 memory subsets comparing the different peptide pools (Fig. 4E). For IL-2 there was a significantly higher frequency of effector memory T cells ($p = 0.0007$ and 0.03) and a lower frequency of $T_{EMRA}$ (CD45RA + CCR7 + ) T cells ($p = 0.001$ and 0.02) in response to PC85 compared to the PC71 and MTB300 respectively (Fig. 4F). The PC85 stimulation also resulted in a significantly higher frequency of effector memory TNFα-producing cells ($p = 0.0003$ and 0.008) and a lower frequency of central memory cells ($p = 0.04$ and 0.02) than PC71 and MTB300, respectively (Fig. 4G). This finding suggests distinct differentiation of the CD4 memory subsets in response to epitope pools representing different functional categories.

## An ATB-specific T cell epitope pool that can distinguish ATB from healthy IGRA+ participants

The data for both the proteome-wide screen and the TB vaccine and IGRA antigens suggests that certain antigens and epitopes are differentially recognized in ATB vs. healthy IGRA+ individuals. Accordingly, we next tested whether reactivity to these "ATB-specific epitopes," could differentiate individuals with ATB at diagnosis, as a cohort more relevant for diagnostic purposes vs. IGRA+ and IGRA- healthy controls. Peptides that were exclusively recognized by participants with ATB (mid-treatment) as compared to our previous studies in IGRA+ participants were pooled into an ATB-pool (ATB116) consisting of 116 peptides (Supplementary Data 1). Reactivity to

MTB300 (constructed based on reactivity in IGRA+ participants) was utilized as a comparator, and none of the peptides in MTB300 overlap with ATB116.

IFNγ response was determined in a Sri Lankan cohort, including 24 patients with ATB (recruited at diagnosis), 25 IGRA + , and 43 IGRA- participants. ATB116 stimulation resulted in significantly higher IFNγ in patients with ATB compared to both IGRA+ and IGRA- controls (Fig. 5A), with 63% of ATB responding compared to 8% of IGRA+ and 16% of IGRA- participants. The frequency of participants responding to MTB300 was similar between the cohorts. MTB300 could not discriminate between ATB and IGRA + , but showed a significantly higher magnitude of reactivity in IGRA+ compared to IGRA- (Fig. 5A). MTB300 contains epitopes that are conserved in nontuberculous mycobacteria, which have been shown to correlate with reactivity observed in IGRA- participants[44]. Similar results were obtained when responses were measured by ICS rather than Fluorospot (Fig. 5B). The IFNγ, IL-2, and TNFα cytokine production was determined in 9 participants with ATB (at diagnosis) and 9 IGRA- participants from Sri Lanka. The frequency of IFNγ and TNFα-producing CD4 cells against the ATB116 was significantly higher in ATB compared to IGRA- participants (Fig. 5B). No differences were observed between these cohorts in response to MTB300.

These results were independently confirmed in an independent cohort from the Republic of Moldova of ATB (at diagnosis and mid-treatment), IGRA + , and IGRA- participants (household contacts of an ATB index case). Similar to what was observed in the case of the Sri Lankan cohort, ATB116 stimulation resulted in a significantly higher magnitude of response in patients with ATB, both at diagnosis and mid-treatment (100% responders), compared to IGRA+ and IGRA- controls (13.4% vs. 15.3% responders; Fig. 5C). MTB300 also showed higher magnitude of response in ATB, both at diagnosis and mid-treatment (100% responders) compared to IGRA+ and IGRA- controls (68% vs. 34.6% responders (Fig. 5C).

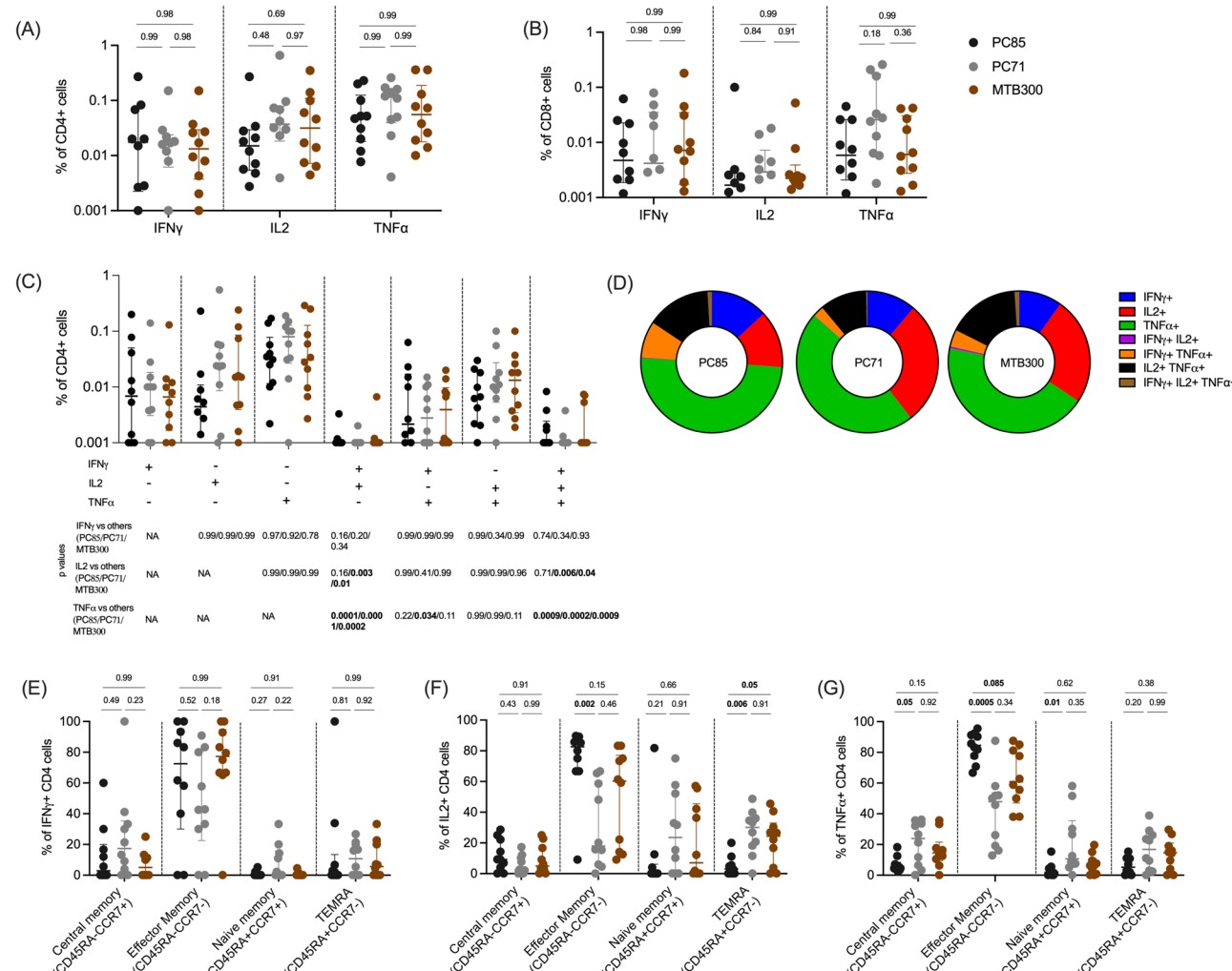

**Fig. 4 | T cell responses specific for different protein categories. A** Frequency of cytokine-producing, IFNγ, TNFα, and IL-2, CD4 T cells in response to PC85, PC71, and MTB300. **B** Frequency of cytokine-producing, IFNγ, TNFα, and IL-2, CD8 + T cells in response to PC85, PC71, and MTB300. **C** Percentage pool-specific IFNγ, TNFα, and IL-2 production by CD4 T cells expressing each of the seven possible combinations. **D** Pie charts representing single, dual and triple cytokine producing CD4 T cells. Each section of the pie chart represents a specific combination

of cytokines, as indicated by the color. **E**−**G** Proportion of CCR7 + CD45RA- (central memory), CCR7-CD45RA- (effector memory), CCR7 + CD45RA+ (naïve), and CCR7-CD45RA+ (T_EMRA) T cells for each peptide pool. **E** IFNγ, (**F**) IL-2, (**G**) TNFα. (**A**−**C**, **E**−**G**) Each point represents one participant from the Peruvian cohort (mid-treatment), $n = 10$, median ± interquartile range is shown. Kruskal Wallis with Dunn's multiple comparison test (two-sided) was used for comparison. Source data are provided as a Source data file.

Next, we calculated the sensitivity and specificity for ATB116 to explore its diagnostic potential. The sensitivity was calculated as the percentage of the participants with ATB responding to ATB116 out of the total ATB cohort. The specificity was calculated as the percentage of IGRA +/− who did not respond to ATB116 out of the total IGRA +/− participants. ATB116 demonstrated a high sensitivity of 62.5% and specificity of over 80% in distinguishing patients with ATB from those who were IGRA+ and IGRA- in the Sri Lankan cohort. Similarly, the sensitivity was 100%, and the specificity was over 80% in the Moldovan cohort differentiating ATB from IGRA+ and IGRA- participants. (Table 1). The diagnostic potential by ROC curve analysis also demonstrated good predictive performance for ATB116 in distinguishing ATB from both IGRA+ and IGRA- in both cohorts (Fig. 6). In the Sri Lanka cohort, the area under the curve (AUC) was 0.83 (Fig. 6A) for distinguishing ATB from IGRA+ individuals and 0.77 (Fig. 6B) for distinguishing ATB from IGRA- individuals. Similarly, in the Moldovan cohort, the AUC was 0.93 (Fig. 6C) for distinguishing ATB from IGRA+ individuals and 0.91 (Fig. 6D) for distinguishing ATB from IGRA- individuals. These findings highlight the diagnostic potential of ATB116 as a useful tool in identifying cases of active tuberculosis.

## The ATB-specific T cell response changes during treatment for ATB
Finally, to investigate the changes in T cell response during treatment, we followed 7 participants in Sri Lanka from diagnosis to mid-treatment and end of treatment. We analyzed the proportion of cytokine-producing CD4 T cells after pool stimulation. In this longitudinal cohort, we observed a significant decrease in ATB116-specific IFNγ responses between visit 1 (at diagnosis) and visit 2 (mid-treatment) (Fig. 6A). At visit 3 (end of treatment), about half the cohort had a further decrease in their response, whereas in the other half, the response against ATB116 increased again. The response against MTB300 did not change between visits 1 and 2 but was significantly decreased compared to visits 2 and 3 (Fig. 7A). The ATB116-specific IL-2 response increased between visits 1 and 3 (Fig. 7B). The response for MTB300 remained the same. Finally, the ATB116-specific TNFα response increased at visit 2 compared to both visits 1 and 3 (Fig. 7C). Again, there was no difference in the response against MTB300. These results indicate that the response against ATB116 changes during treatment for ATB and is differentially affected depending on the specific cytokine measured.

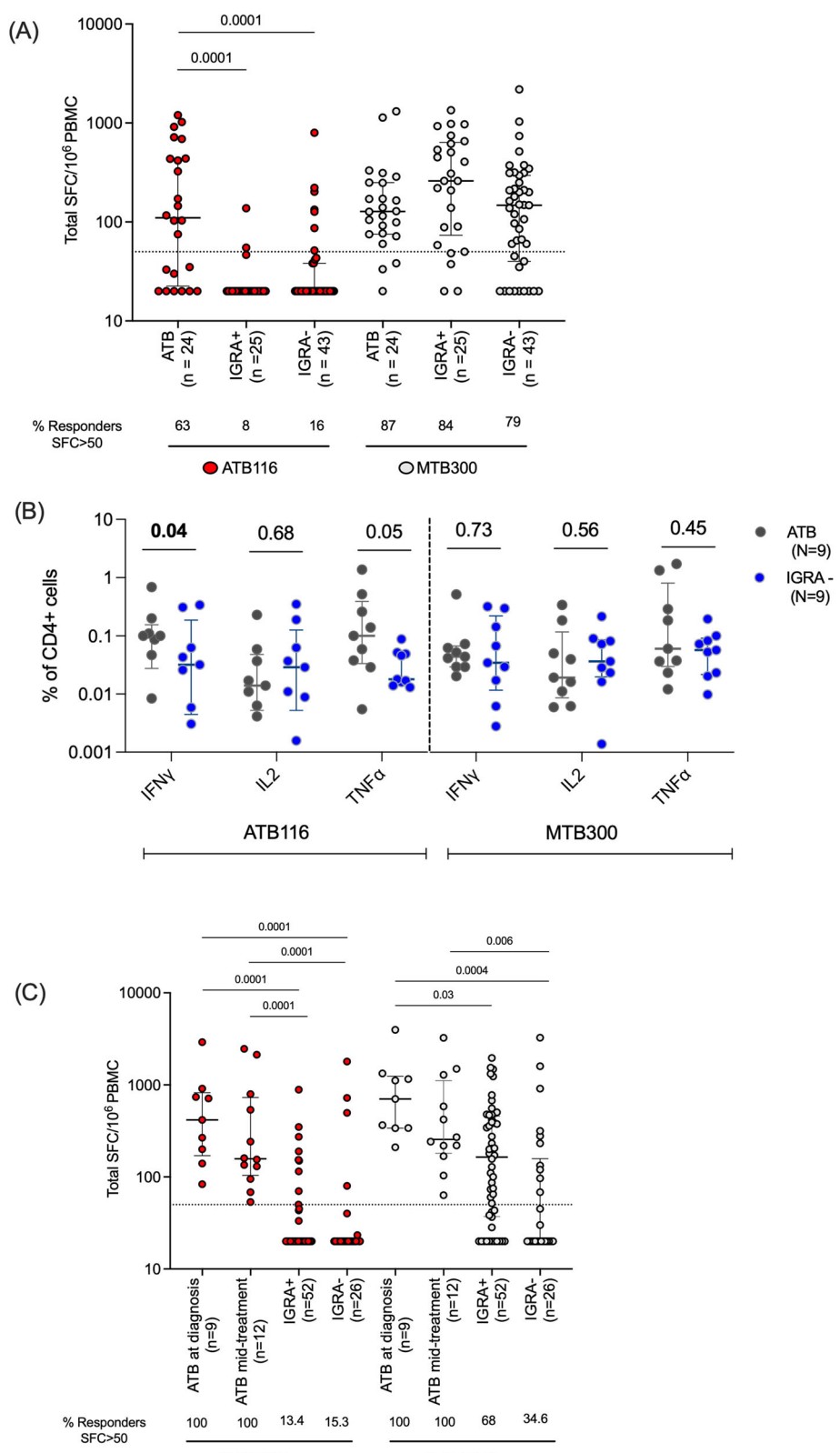

**Fig. 5 | Cytokine response against the ATB-specific peptide pool, ATB116.**
**A** Magnitude of response (total SFC for IFNγ) against ATB116 (red) and MTB300 (white) in ATB (at diagnosis; $n = 24$), IGRA+ ($n = 25$), and IGRA- ($n = 43$) from Sri Lanka. **B** Frequency of cytokine-producing, IFNγ, TNFα, and IL-2, CD4 T cells in response to ATB116 and MTB300 in ATB (at diagnosis, $n = 9$; grey) and IGRA- ($n = 9$; blue) participants. **C** Magnitude of response (total SFC for IFNγ) against ATB116 (red) and MTB300 (white) in ATB (at diagnosis; $n = 9$), ATB (mid-treatment; $n = 12$),

IGRA+ ($n = 52$), and IGRA- ($n = 26$) household contacts from Moldova. **A**–**C** Each dot represents one participant, median ± interquartile range is shown. Kruskal Wallis with Dunn's multiple comparison tests (two-sided) was used to compare IFNγ response among ATB, IGRA + , and IGRA- (**A**, **C**). Mann Whitney test (two-sided) was used to compare cytokine levels in ATB and IGRA- (**B**). **A**, **C** The dashed line indicates the cut-off used for a positive response (50 SFC/$10^6$ PBMCs). Source data are provided as a Source data file.

## Discussion

Here, we report the first proteome-wide identification of *Mtb*-derived T cell epitopes in a cohort of patients with ATB. We have defined the epitopes from a library of over 21,000 peptides, with an in-depth investigation of the antigens in TB vaccine candidates. The lack of understanding of the broad range of *Mtb* antigens that may elicit a differential T cell response between individuals with various stages of *Mtb* infection is a bottleneck in developing diagnostic assays and vaccines. Many studies have focused on the identification of novel antigens which can be utilized for better diagnostic assays and vaccine development[21,45–48]. Despite many efforts, only few antigens, < 20, of the over 4000 ORFs in the *Mtb* genome are included in subunit vaccine candidates[49–53].

Comparing the results described here with those from our previous studies in healthy IGRA+ participants[22] revealed differentially recognized epitopes and antigens between ATB and IGRA+ participants. This differential recognition of antigens can be explained by *Mtb*'s expression of infection stage-specific antigens, as previously reported[54]. For instance, during *Mtb* infection, *Mtb* alters its metabolic state from active replication to slow or nonreplicating, accompanied by changes in the gene expression profile and thus protein expression and antigens available to the immune system[55]. The differential expression and availability of antigens at distinct stages of TB infection might exhibit distinct patterns of differentiation and restricted capacities to mediate protective immunity[56].

The proteome-wide screen in IGRA+ participants revealed three antigenic islands that were immunodominant and related to type 7 secretion systems, with secreted and secretion apparatus proteins being recognized as antigens[22]. Here, we found reactivity against antigens that are part of the antigenic islands, albeit with a more restricted recognition of primarily secreted proteins. The underlying cause of this is unclear, but it could be due to *Mtb*'s differential expression of proteins in different stages of TB infection. Our studies also highlighted a hierarchy of responses for the vaccine candidate and IGRA antigens in the two cohorts, with the well-studied antigens, Rv0288, Rv3875, Rv3874, and Rv1196, being the most frequently recognized. This highlights the complexity of *Mtb*-specific T cell responses and suggests the need for vaccine strategies targeting many different antigens.

**Table 1 | Diagnostic potential of ATB116 distinguishing patients with ATB from IGRA+ and IGRA- participants**

|  | Cohort | Sensitivity (95% CI) | Specificity (95% CI) |
|---|---|---|---|
| Sri Lanka | ATB vs IGRA+ | 62.5 (42.7–78.8) | 92 (75.0–98.6) |
|  | ATB vs IGRA- |  | 83.7 (70.0–91.8) |
| Moldova | ATB vs IGRA+ | 100 (84.5–100) | 84.6 (72.5–92.0) |
|  | ATB vs IGRA- |  | 80.9 (66.5–93.9) |

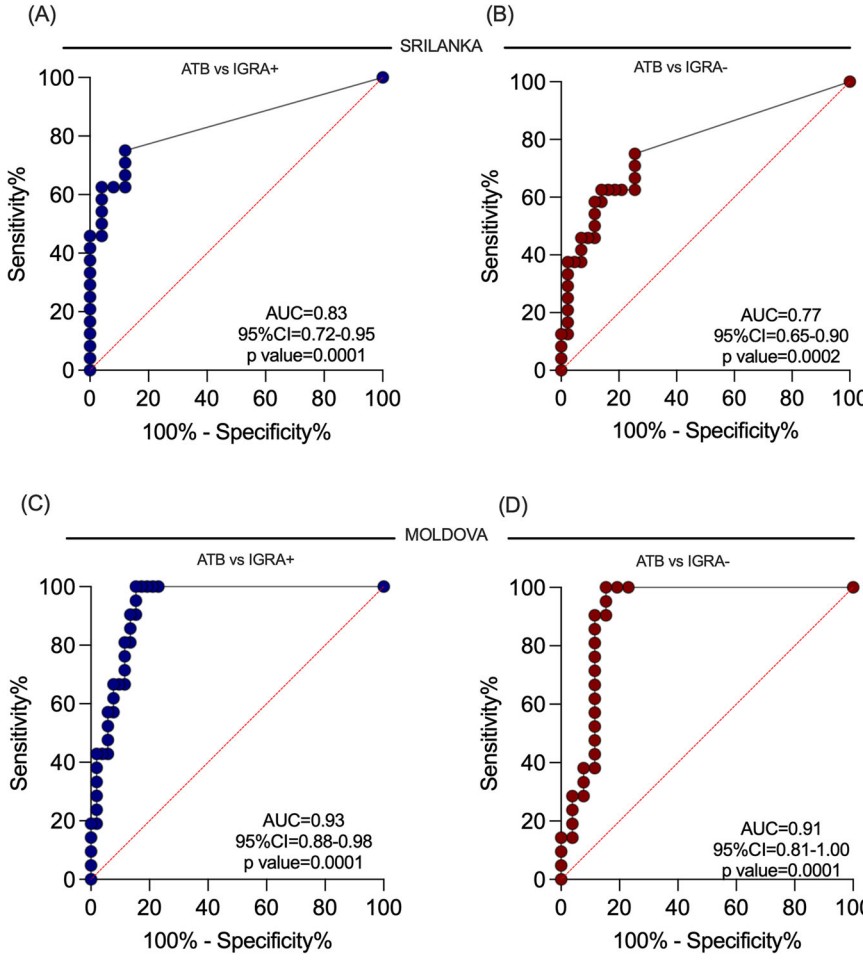

**Fig. 6 | The diagnostic potential of ATB116.** Receiver Operating Characteristic (ROC) curves for Sri Lanka (**A**, **B**) and Moldova (**C**, **D**). ATB vs. IGRA+ (**A**, **C**) and ATB vs. IGRA- (**B**, **D**). The x-axis shows the false positive rate (100%-specificity%), and the y-axis shows the true positive rate (sensitivity %). Each point on the curve represents a different threshold value used to calculate the true and false positive rates. The area under the curve (AUC) is shown for predictive performance, including the confidence interval (CI) and associated two-sided *p*-value. Source data are provided as a Source data file.

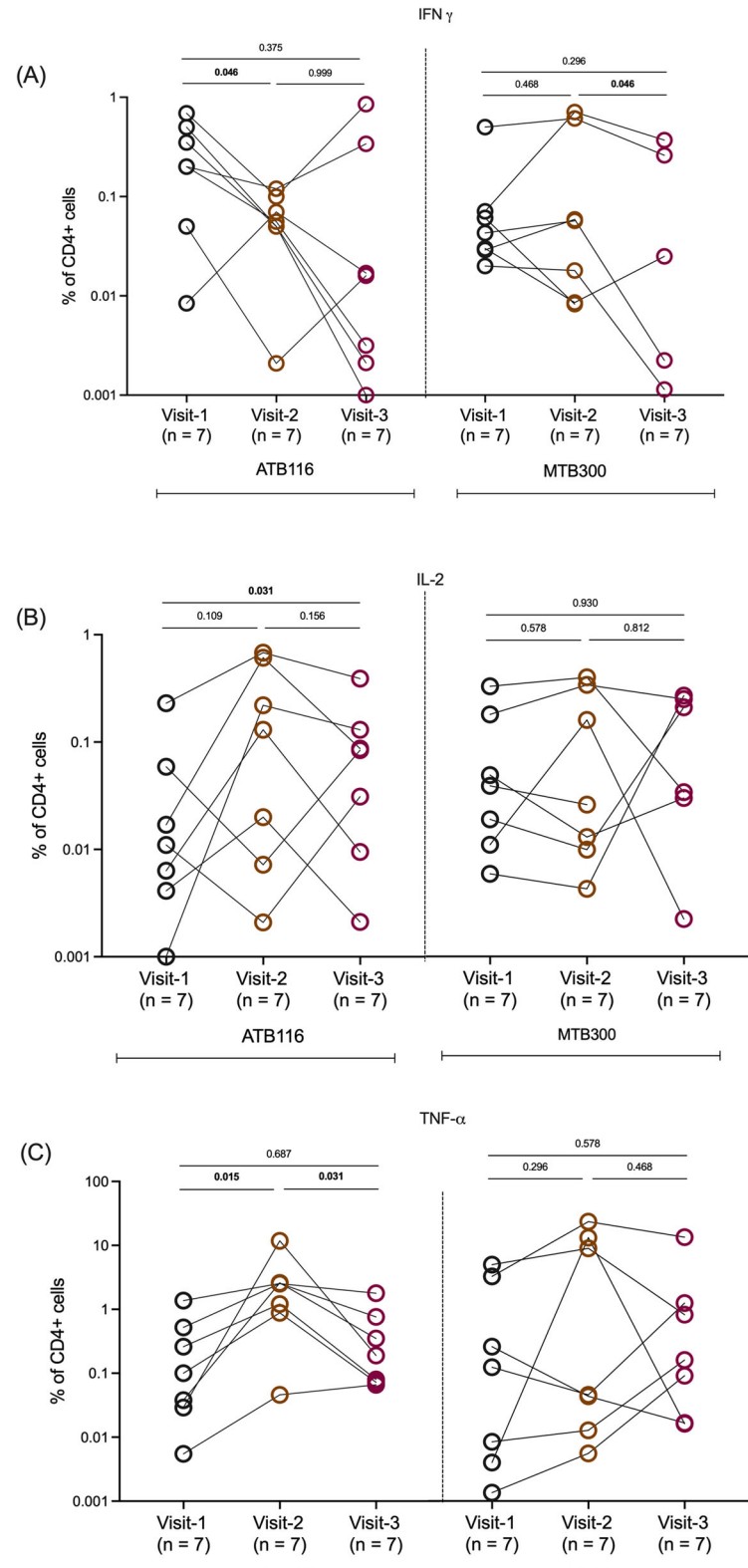

**Fig. 7 | Characterization of ATB116- and MTB300-specific CD4 T cell responses.**
**A–C** Frequency of cytokine-producing, IFNγ (**A**), IL-2 (**B**), and TNFα (**C**), CD4 T cells in response to ATB116 and MTB300 in longitudinal samples from patients with ATB (visit 1; at diagnosis, visit 2; mid-treatment, visit 3; end of treatment, *n* = 7). Each point represents one participant, median ± interquartile range is shown. Wilcoxon signed rank test (two-sided) was used. Source data are provided as a Source data file.

The previous work identifying epitopes from IGRA+ participants led to the development of a peptide "megapool", MTB300, which allows for capturing diverse *Mtb*-specific responses irrespective of the HLA alleles expressed in the population[42]. Using the same multi-epitope approach here, we defined a new megapool, ATB116, which contains epitopes that are not included in MTB300 and have only been found to be reactive in participants with ATB. Specifically, ATB116 could discriminate ATB at diagnosis and mid-treatment from control participants who were IGRA+ and IGRA-, in geographically diverse cohorts from Sri Lanka and Moldova. There was a higher frequency of reactivity against ATB116 in IGRA+ and IGRA- controls from Moldova compared to Sri Lanka. This could be because the Moldovan cohort of IGRA + /- are household contacts within 6 months of a primary ATB index case and have thus been recently exposed to *Mtb*. Further studies will determine whether this reactivity is simply a measure of recent exposure or whether these individuals will develop active TB in the future. In addition, future studies will determine whether responses against the ATB116 epitopes are protective or pathological. We observed a decrease in IFNγ response upon treatment, suggesting that antigen expression levels change during treatment as it takes effect. This and the ATB-specific recognition may indicate that these peptides are not protective. However, no firm conclusions can be drawn from the results presented here. No IL-17 was detected in the individuals with ATB mid-treatment, but this does not rule out differences between IFNγ and IL-17 at diagnosis of ATB, where we have not measured IL-17. Contradictory findings have been reported for the role of polyfunctional CD4 + T cells in relation to host containment of *Mtb* infection. On one hand, stronger polyfunctional *Mtb*-specific CD4 + T cell responses have been observed in adults with acid-fast (AFB) smear negative sputum samples and latent TB infection (LTBI) compared to those with AFB smear positive tuberculosis (TB)[57–59]. Consistent with our observations, these studies have also reported a greater proportion of T cells producing TNFα. IFNγ, IL-2, and TNFα responses were similar when cells were stimulated with the peptide pools corresponding to different protein categories. In the longitudinal samples, IFNγ decreased from diagnosis to the second time point 2 m post diagnosis, TNFα increased, and IL-2 showed a trend for an increase in response to ATB116. Successful TB treatment, which reduces the bacterial load, is also associated with significant increases in the proportion of polyfunctional CD4 + T cells. On the other hand, there are studies demonstrating a positive correlation between polyfunctional CD4 + T cell responses and increased bacillary load, showing stronger responses in adults with TB compared to those with LTBI and healthy household contacts[60,61]. These contradictory results highlight the limitations of correlative studies in determining whether the polyfunctionality of CD4 + T cells plays a causal role in the immune control of the pathogen or reflects the underlying bacterial burden. We have previously described a gene signature in individuals with latent TB at risk of developing active TB, which overlaps with a signature from ATB[62]. This gene signature approach, together with ATB116-specific reactivity, may be useful for identifying individuals at risk of developing active TB. The differences in sensitivity observed between the Moldovan and Sri Lankan cohorts may reflect differences in the infecting *Mtb* strains, differential exposure to NTM and other environments or infections, and potential host factors. Taken together, further exploration of the ATB116 pool for its diagnostic potential across the spectrum of *Mtb* infection is warranted. Ideally, prospective studies to investigate whether reactivity to ATB116 can predict progression to ATB. In addition, ATB116, or versions thereof, may be useful as a whole blood assay in areas with high levels of IGRA positivity where there is a need to selectively identify ATB.

We could not detect a functionally distinct immune response in terms of multifunctionality. The majority of the responding T cells, irrespective of the functional categories, produced a single cytokine, with the majority producing TNFα. Some differences were observed regarding memory phenotype, where the cell wall and cell processes category had a higher frequency of effector memory T cells than the other categories. This suggests that T cell responses against these epitopes result in increased differentiation of the responding memory subset[63]. Overall, we observed that epitope-specific T cells were predominantly CD45RA-CCR7- effector memory cells, followed by CD45RA- CCR7 +, which agrees with previous studies in patients with ATB[64–67]. Taken together, it has previously been shown that differentiation of T cells towards later-stage effector memory during ongoing antigen expression primarily favors the expression of IFNγ and/or TNFα[63,68], as observed here. Ongoing studies are investigating the ATB116 and other epitope-specific T cell phenotypes and subsets in more detail, including the involvement of Th1*[22,62,69,70].

In conclusion. this study addressed a critical knowledge gap in understanding the diversity of *Mtb* antigens capable of eliciting differential T cell responses during various stages of infection by identifying *Mtb*-derived T cell epitopes in patients with active tuberculosis (ATB) using a proteome-wide library with over 21,000 peptides, including TB vaccine candidate antigens. As a result, a novel multi-epitope "ATB116" pool was created, capable of discriminating ATB from IGRA+ and IGRA- individuals, offering the potential use in diagnostic tests and measuring *Mtb*-specific immune responses.

## Methods

### Study approval

All participants provided written informed consent for participation in the study. Ethical approval was obtained from the institutional review boards at La Jolla Institute for Immunology (LJI; Protocol Numbers: VD-090, VD-143, VD-175), Universidad Peruana Cayetano Heredia (66754), Phthisiopneumology Institute (CE-3/2018), University of California San Diego (180068), and University of Colombo for General Sir John Kotelawala Defense University, Sri Lanka (EC18-122, EC15-094). Results for IGRA+ individuals from San Diego, USA (Fig. 2D) and the Western Cape region, South Africa (Fig. 3B) are included here for comparison purposes. They have been reported previously; San Diego cohort[22], and South African cohort[42].

### Study participants

We recruited a total of 184 participants over 18 years of age for this study from three different cohorts, UPCH in Peru, the General Sir John Kotelawala Defense University in Sri Lanka, and the Phthisiopneumology Institute in the Republic of Moldova. From Peru, 21 participants with ATB who were mid-treatment (3-4 months post-diagnosis) were recruited from 2012 to 2013. The cohort consisted of 62% males and 38% females. From Sri Lanka, a cohort of patients with ATB (at diagnosis, $n = 24$; 96% males and 4% females), IGRA+ individuals ($n = 25$; 48% males and 52% females), and IGRA- individuals ($n = 43$; 39% males and 60.5% females) were recruited from 2019 to 2022. A subset of the patients with ATB ($n = 7$) was followed longitudinally from the time of diagnosis until the end of treatment. They provided blood samples at diagnosis, 2 months post-diagnosis, and 6 months post-diagnosis. From Moldova, a cohort of patients with ATB (at diagnosis, $n = 9$), ATB (mid-treatment, $n = 12$), IGRA+ individuals ($n = 52$), and IGRA- individuals ($n = 26$) were recruited from 2018-2022 (no biological sex information available). The IGRA+ and IGRA- individuals were household contacts of a patient with active TB (e.g., an "index case"). They provided blood samples up to 6 months after the index case received their ATB diagnosis.

Healthy participants were classified into IGRA+ (i.e., Latent TB infection) and IGRA- groups based on IGRA tests (QuantiFERON-TB Gold Plus, Cellestis and/or T-spot.TB, Oxford Immunotec). Individuals with ATB had symptomatic pulmonary TB, diagnosed by a positive GenXpert (Cepheid, Inc.), positive sputum smear, and/or a positive culture.

Participants with pulmonary TB recruited in Peru were aged 18-50, had documented culture confirmed TB from sputum, and were currently 3–4 months post-diagnosis of ATB. They responded well to treatment and had regained any weight lost due to the infection. Individuals with MDR or XDR-TB, and those diagnosed HIV, HBV or HCV infection were excluded, as well as patients with significant systemic diseases, including, for example, diabetes, renal disease, liver disease, uncontrolled hypertension, and malignancy. They provided 100 ml leukapheresis samples.

## PBMC isolation and thawing
PBMCs were obtained by density gradient centrifugation (Ficoll-Hypaque, Amersham Biosciences) from leukapheresis or whole blood samples, according to the manufacturer's instructions. The PBMC processing at the site in the Republic of Moldova used SepMate tubes (StemCell). Cells were resuspended in FBS (Gemini Bio-Products) containing 10% DMSO (v/v, Sigma) and cryopreserved in liquid nitrogen.

Cryopreserved PBMC were quickly thawed by incubating each cryovial at 37 °C for 2 min, and cells transferred to cold medium (RPMI 1640 with L-glutamin and 25 mM HEPES; Omega Scientific), supplemented with 5% human AB serum (GemCell), 1% penicillin streptomycin (Life Technologies), 1% glutamax (Life Technologies) and 20 U/ml benzonase nuclease (MilliporeSigma). Cells were centrifuged and resuspended in complete RPMI medium to determine cell concentration and viability using trypan blue. All samples had a > 60% viability.

## Peptide screening and peptide pool preparation
The present study screened a total of 21,220 peptides. These peptides include the same library that was screened in IGRA+ participants of 20,610 *Mtb* peptides (2 to 10 per ORF, average 5), including 1660 variants not totally conserved amongst the selected *Mtb* genomes: five complete *Mtb* genomes (CDC1551, F11, H37Ra, H37Rv and KZN 1435) and sixteen draft assemblies[22]. Along with these peptides, 610 peptides were also selected, which include 93 peptides that are not found in *Mtb* but present in the *Mycobacterium bovis* BCG strains Mexico, Tokyo 172, and Pasteur 1173P2, and 517 peptides that are 15-mers overlapping by ten amino acids spanning the entire sequence of 12 TB vaccine candidate and IGRA antigens. The vaccine candidates included ID93: GLA-SE (Rv3619c, Rv3620c, Rv1813, and Rv2608), H1:IC31 (ESAT-6 and Ag85B), H4:IC31 (Ag85B, TB10.4), H56:IC31 (Ag85B, ESAT-6, and Rv2660c), M72/AS01E (Mtb32A, PPE18), and three candidates with Ag85A alone (Ad5 Ag85A, ChAdOx1-85A/MVA85A, and MVA85A). The IGRA antigens include ESAT-6, which is also a vaccine candidate antigen, and CFP10.

The peptides were synthesized as crude material on a small (1 mg) scale by Mimotopes (Australia). The peptides were solubilized using DMSO at a concentration of 20 mg/ml. The peptides were pooled into peptide pools. The previous peptide library was pooled into 1036 peptide pools of about 20 peptides each (average 19.9 ± 0.5). The 93 non-*Mtb* peptides were pooled into 5 peptide pools. The 15-mer peptides overlapping by 10 amino acids spanning the TB vaccine candidate and IGRA antigens were pooled into 26 peptide pools. Thus, from the 21,220 total peptides, a total of 1067 peptide pools were made.

Individual peptides were mixed in equal amounts after being dissolved in DMSO for the megapools (PC85, PC71, ATB116, and MTB300), as described previously[42]. Each peptide pool was then placed in a lyophilizing flask and subjected to lyophilization for 24 h. The resulting semi-solid product was dissolved in water, frozen, and lyophilized again until only solid product remained. This process was repeated several times until only solid product remained after lyophilization. Finally, the peptide pool was re-suspended in DMSO at a higher concentration per peptide (0.7 mg/ml per peptide) than before lyophilization, to reduce the concentration of DMSO in the assays.

## Ex vivo IFNγ and IL-17 fluorospot assay
IFNγ and IL-17 production was measured by a Fluorospot assay with all antibodies and reagents from Mabtech (Nacka Strand, Sweden). Plates were coated overnight at 4 °C with an antibody mixture containing mouse anti-human IFNγ (clone 1-D1K) and mouse anti-human IL-17 (clone MT44.6). Briefly, $2 \times 10^5$ PBMCs were added to each well of pre-coated Immobilion-FL PVDF 96-well plate in the presence of peptide pools at a concentration of 2 µg/ml, individual peptides at 5µg/ml, PHA at 10µg/ml (positive control) and media containing DMSO (amount corresponding to percent DMSO in the pools/peptides, as a negative control). Plates were incubated at 37 °C in a humidified $CO_2$ incubator for 20–24 h. All conditions were tested in triplicate, except the negative control, which was tested in six individual wells. After incubation, plates were developed according to the manufacturer's instructions. Briefly, cells were removed and wells were washed with PBS/0.05% Tween 20 using an automated plate washer. After washing, an antibody mixture containing anti-IFNγ (7-B6-1-FS-BAM) and anti-IL-17 (MT504-WASP) prepared in PBS with 0.1% BSA was added to each well and plates were incubated for 2 h at room temperature. The plates were again washed and incubated with diluted fluorophore-conjugated anti-BAM-490 and anti-WASP-640 antibody for 1 h at room temperature. Finally, the plates were washed and incubated with a fluorescence enhancer for 15 min, blotted dry and fluorescent spots were counted by computer-assisted image analysis (IRIS Fluorospot reader, Mabtech, Apex version 1.1.59.128, Sweden).

Each pool or peptide was considered positive compared to the background that had an equivalent amount of DMSO based on the following criteria: (i) 20 or more spot-forming cells (SFC) per $10^6$ PBMC after background subtraction, (ii) a greater than 2-fold increase compared to the background, and (iii) $p < 0.05$ by student's t-test or Poisson distribution test when comparing the peptide or pool triplicates with the negative control. The response frequency was calculated by dividing the number of participants responding by the no. of participants tested. The magnitude of response (total SFC) was calculated by summation of SFC from responding participants.

## Intracellular cytokine staining assay
PBMC at $1 \times 10^6$ per condition were stimulated with peptide pools (2 µg/ml) for 18–20 h in complete RPMI medium at 37 °C with 5% $CO_2$. PBMCs incubated with DMSO at the percentage corresponding to the amount in the peptide pools were used as a negative control to assess nonspecific or background cytokine production, and anti-CD3/CD28 (1 µg/ml; OKT3 and CD28.2) stimulation was used as a positive control. For Chemokine receptor staining, antibodies were added during the stimulation. After 18 h, 2.5 µg/ml each of BFA and monensin was added for an additional 5 h at 37 °C. Cells were then harvested and incubated in a blocking buffer containing 10%FBS and 1 µg/mL Human Fc block (BD Biosciences, USA) for 20 min at 4 °C. Next, cells were stained using fixable live/dead stain for 20 min at room temperature and then stained with surface-expressed antibodies; (Supplementary Table 1) diluted in FACS buffer and 1X Brilliant Stain Buffer (BD Biosciences, USA) for 20 min at room temperature. Cells were permeabilized and fixed for intracellular cytokine staining using cytoperm fixation buffer (Biolegend) for 20 min at room temperature. After incubation, cells were stained for cytokines (IFNγ, IL-2 and TNFα) for 20 min at room temperature. Samples were acquired on a ZE5 cell analyzer (BioRad). Frequencies of CD4 or CD8 T cells responding to each peptide pool were quantified by determining the total number of gated CD4 or CD8 and cytokine-producing cells and background values subtracted (as determined from the negative control) using FlowJo X Software. Combinations of cytokine-producing cells were determined using Boolean gating. The lower limit of detection for the frequency of cytokine-producing CD4 T cells after background subtraction was set to 0.001%.

## Statistical analysis

Statistical analyses were performed using GraphPad Prism software (GraphPad Software, Inc., San Diego, CA, USA, version 9.2). Data is shown as median with interquartile range. Non-parametric test was applied after checking for normality. A two-tailed Mann-Whitney U test was used for comparison between two groups. Wilcoxon signed-rank tests were used for the comparison of cytokine-producing cells in longitudinal samples from patients with ATB. $p$-value $\leq 0.05$ was considered significant. No statistical method was used to predetermine the sample size. No data were excluded from the analysis. The experiments were randomized. The investigators were blinded to cohort allocation during the experiment setup and unblinded during analysis.

## Reporting summary

Further information on research design is available in the Nature Portfolio Reporting Summary linked to this article.

## Data availability

The identified epitopes from the proteome-wide screen in participants with ATB have been submitted to IEDB (Immune Epitope Database; www.iedb.org) and can be found under the submission ID: 1000914. H37Rv genome was used as reference genome (GenBank accession number NC_000962) for comparing protein categories between identified antigens and overall in the *Mtb* genome. The data generated in this study are provided in the Supplementary information and Source Data file. Source data are provided with this paper.

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

## Author contributions

C.S.L.A., A.S., and B.P. participated in the design and direction of the study. S.P., J.M., C.C., and C.S.L.A. performed and analyzed the experiments. M.S., R.H.G., N.C., V.C., D.G.C., A.C., T.R., J.S.B.P., T.C., B.G., and A.D. DS, recruited participants, performed clinical evaluations, and isolated PBMCs. C.S.L.A. and S.P. wrote the manuscript. All authors read, edited, and approved the manuscript before submission.

## Funding

This work was supported by the National Institutes of Health contract HHSN272200900044C (to A.S.), 75N93019C00067 (to B.P. and C.S.L.A.), and R01 AI137681 (to A.C.). The funders had no role in study design, data collection, analysis, decision to publish, or manuscript preparation.

## Competing interests

The authors declare no competing interests.
