## [Peer Review File · Nature Communications]

Identification of differentially recognized T cell epitopes in the spectrum of tuberculosis infectionReviewers' Comments:

Reviewer #1:

Remarks to the Author:

In this study, the authors use a previously established peptide library of 20,610 peptides as well as 1127 additional peptides obtained eg. from different TB vaccine candidates and IGRA antigens, to study the T cell reactivity in 21 Peruvian patients with active pulmonary tuberculosis (ATB). The main readouts were IFN γ ELISPOT in response to different peptide pools, as well as intracellular cytokine staining and flow cytometry for a subgroup of ATB. 116 ATB-specific peptides were also selected and used to determine the IFN γ response in a Sri Lankan cohort consisting of 24 ATB patients, 25 IGRA+ and 43 IGRA- individuals, as well as a Moldavian cohort consisting of 21 ATB patients, 20 IGRA+ and 10 IGRA- individuals. Blood samples obtained from the different patient cohorts were collected at different time-points after start of anti-TB treatment. The conclusion is that epitopes and antigens are differentially recognized depending on the stage of infection, and may be used to develop diagnostics, new vaccine candidates and correlates of protection.

Although, I appreciate and understand the cumbersome effort to perform these analyses, the study outline and analyses are limping a bit and as a reader it is difficult to obtain a clear overview of the different sub-analyses and how the results from these bring significant added value to current knowledge. The large peptide library was generated 10 years ago and investigated using samples from 28 individuals with latent TB as well as 28 healthy controls. It appears as the current study use a similar approach, but with ATB patients instead. Some of the LTBI data are also included for comparisons. It is unclear when and how the different cohorts were recruited, and if the study was systematically designed to include ATB patients "mid-treatment" and if so, what the benefit of that would be. From this study it is not clear what represents immune correlates of protection or if it practical and relevant to use the ATB116 peptide pool to differentiate ATB from LTBI or healthy controls.

Specific comments:

1. The rationale for using PBMC samples from ATB patients "mid-treatment" needs to be motivated. Already after intensive-phase treatment 2 month after start of treatment, the Mtb load and secreted antigens may be very different compared to ATB patients at diagnosis. This is of particular importance as the longitudinal flow data in Figure 6, suggest that the CD4+ cytokine responses are quite different comparing visit 1, 2 and 3. Then the IFN γ responses in the Sri Lankan cohort included ATB patients recruited at diagnosis and Moldovan ATB patients recruited at diagnosis and mid-treatment. I'm also curious how the sample sizes were determined. In the MM section it is explained that 164 Peruvian patients were recruited; so why were only 21 included in the peptide analyses? How were these 21 patients selected among the total of 164? And how would the data presented in Figure 1 change if a higher number of patients would have been included? Overall, please add the time frame and years in which the different patient cohorts were recruited. A supplementary table to clarify this may also help.
2. For the ICS in Figures 4 and 5, how were the sub-group of n=10 patients chosen? Why not analyse all 21 ATB patients? The font size in Figure 4 is too small and should be increased. Furthermore, it is not appropriate to use a Mann-Whitney test to compare groups of three (PC85, PC71, MTB300) but a Kruskal-Wallis test should be used. Similarly, pair-wise comparisons using Mann-Whitney in Figure 5 is not correct.
3. For the sensitivity and specificity calculations in Table 1, ROC curves should be added to illustrate the data. The fact that the Sri Lankan cohort display 62,5% sensitivity and the Moldovan cohort 65-70% specificity should be discussed. How applicable are these results for diagnostic purposes?
4. As most of these data are build on the IFN γ response to different peptide pools (IL-17 was undetectable?) it would be good if the authors discussed IFN γ , TNF α and IL-2 as representative correlates of protection and any new literature related to these markers. Is it good to have higher

proportions of effector memory cells expressing IFN γ and TNF- α ? Are these T cells likely to be protective, and if so how?

5. In Figure S1, please show the viability staining of the CD3 $^+$ T cells. What markers were included in the DUMP; CD19 and CD14 only? The DUMP/CD3 plot looks odd.

6. For the flow cytometry data, do the authors have the possibility to perform some type of unsupervised analyses of their data such as UMAP to study the clustering of T cell subsets and the expression of different cytokines? I believe this would strengthen the data significantly.

Reviewer #2:

Remarks to the Author:

Comments to the authors:

The authors describe T-cell responses against 21.000 peptides derived from MTB with the aim to differentiate active tuberculosis (ATB) versus healthy individuals. The screening is performed in PBMCs from cohorts gauging IFN- γ and IL-17 responses. More detailed analysis is performed using a combination of intracellular cytokine staining (ICS) and cell surface marker analysis, e.g. CD45RA / CCR7 in order to defined T-cell subsets (precursor, memory, terminally differentiated T-cells). The authors suggested that T-cell reactivity directed against distinct antigens, such as ATB116 may be able to differentiate between patients with active TB and others.

Specific comments:

1. the participants are addressed as ‘ ‘ATB patients’ ’ - I would suggest to rephrase patients with ATB, i.e. so it reflects that it is an individual with a disease and not that the disease defines the individual.

2. lane 221: TNF alpha production was increased in contrast to IFN, similarly to an older paper from S Kaufmann in Nature. TNFalpha is the cytokine which can still be produced by exhausted immune cells, even if IL-2 and/or IFN- γ does not happen: is the TNFalpha production associated with terminally differentiated immune cells or T-cells with exhaustion markers ? Please comment. Could TNFa a more useful cytokine as compared to gauging other cytokines, in gauging biologically relevant T-cell responses against defined MTB antigens ? Pls comment.

3. lane 113 and figure 1: IFN and IL-17 production was tested in the screening. I may have missed it, yet I had to a hard time to spot IFN versus IL-17 responses. If I miss that point, the readers may miss it as well.

3.1. Pls comment on IFN- γ versus IL-17 production in response to the target antigens.

3.2. Would different patterns emerge as different qualities of T-cell responses are being analyzed ?

3.3. Are there distinct groups of reactivity defined by high IFN versus high IL-17 or high IL-17 plus IFN γ production in response to candidate peptides ?

3.4. Pls provide an introduction why IFN- γ and IL-17 was chose as the screening

3.5. Were they also commonly shared targets e.g. CMV or EBV included in the screening and would these targets show differences in patients with ATB ? (see comments in point 4 concerning anergy).

4. The testing cohort were 24 patients with ATB from Peru, the blood was harvested 3-4 months after diagnosis

from Sri Lanka and Moldavia: 24 patients with ATB from each study place yet blood was collected at the time of diagnosis.

4.1. Was there an association with high IFN- γ responses and the Peru cohort ? Since T-cells from patients with active TB harvested at the time of diagnosis show often anergy and reduced

immune - reactivity against antigens in general. Please show the responses based on the PBMC harvest (3/4 months versus at the time of diagnosis of ATB).

4.2. Were there differences in cytokine productions in a) PBMCs harvested at the time of diagnosis versus later concerning IFN-gamma and IL-17, and b) in ICS, i.e. examining IL-2, TNFalpha and IFN-gamma ?

4.3. I am confident that ATB does along with at least an Xray. were the study participants stratified based on extent of the disease based on the XRays (more extensive disease is usually associated with increased immune-suppression) ?

4.4. Pls provide data whether TB was sensitive to the conventional antibiotic regimens or XDR or MDR TB (or total-drug resistant).

4.5. is there any bias in the cohort based on different exposures and genetic background: pls show Figure 1 in all its elements based on the three different cohorts (Peru, Sri Lanka, Moldavia). If all looks the same, pls comment and move it in the supplementary data sets.

5. lane 438: the patients who donated 100mL blood underwent a leucapheresis ? This would be very unusual for 100 mL.

6. What is the biological meaning of T-cell reactivity to the target antigen associated with ATB ?

7. What is the clinical consequence and what would be the advantage to run an immunological test (as compared an IGRA, Xray, culture or PCR GenExpert) ?

8. PBMCs after thawing: what was the viability of T-cells AT THE TIME THE FUNCTIONAL ASSAY WAS STARTED. This is important since T-cells from patients with a heavy disease burden usually dont freeze very well - and tend to produce more TNFalpha as compared to other cytokines.

Reviewer #3:

Remarks to the Author:

I have carefully read the paper by Panda et al, and have a number of concerns:

- IL-17 is mentioned in the abstract and methods but not in de the actual data, the data should be described or all mentions should be removed from the manuscript

- The data from the IGRA+ participants to the 1036 peptide pools were previously collected, since the publication is already from 2013 the data must be even older. How do the authors guarantee translatability of these data and is the alignment with the ATB patients in the current paper made, in particular with the direct comparison that is made in Figure 2D and the magnitude of the responses.

- Figure 2D is hard to read and data should be plotted in a different way.

- The data on the 517 peptides from the TB vaccine candidates seems unrelated to the 20.610 peptides described for the rest of the data, also the ATB116 peptides are derived purely from the 20.610 peptides, so what is the added value of these data? Why are they included here? The selection process of the peptides to be taken forward is also not completely clear and the description should be extended. It is mentioned that the ATB116 are non-overlapping with the MTB300 peptides but is it unclear in what stage this has been selected, was that already done before screening or were screening results further tailored to arrive at the 116 peptides?

- Overall, not all ATB patients, either at mid-treatment or at diagnosis, recognize the ATB116 pool, but approximately 67% does. How do the authors envision this to be applied in diagnostic testing? In particular because also the IGRA+/- individuals from Moldova with a recent exposure as household contact seem to have the highest background responses, and those type of individuals would be the population to be tested in diagnostic work up? Wouldn't that decrease the value of ATB116 for diagnosis if recent exposure can induce similar responses? This should be discussed more extensively.

- Figure 4 should be described better, it is unclear which donors were used here, which cohort and which disease state?
- The cytokine production and memory states were determined towards 2 peptide pools (Figure 4), PC71 and PC85, but it is unclear what the relation between those peptide pools and the ultimate ATB116 pool is. Why is MTB300 used as a comparator here and not ATB116?
- Information on ethical permission for the Sri Lankan cohort is missing
- The cytokine data are described in the methods to have a cut off of 0.0001%, which is only 1 cell if starting with 1×10^6 PBMCs... please check if this has been truly applied.

Point-by-point reply to reviewer's comments.

Reviewer #1 (Remarks to the Author):

In this study, the authors use a previously established peptide library of 20,610 peptides as well as 1127 additional peptides obtained eg. from different TB vaccine candidates and IGRA antigens, to study the T cell reactivity in 21 Peruvian patients with active pulmonary tuberculosis (ATB). The main readouts were IFN γ ELISPOT in response to different peptide pools, as well as intracellular cytokine staining and flow cytometry for a subgroup of ATB. 116 ATB-specific peptides were also selected and used to determine the IFN γ response in a Sri Lankan cohort consisting of 24 ATB patients, 25 IGRA+ and 43 IGRA- individuals, as well as a Moldavian cohort consisting of 21 ATB patients, 20 IGRA+ and 10 IGRA- individuals. Blood samples obtained from the different patient cohorts were collected at different time-points after start of anti-TB treatment. The conclusion is that epitopes and antigens are differentially recognized depending on the stage of infection, and may be used to develop diagnostics, new vaccine candidates and correlates of protection.

Although, I appreciate and understand the cumbersome effort to perform these analyses, the study outline and analyses are limping a bit and as a reader it is difficult to obtain a clear overview of the different sub-analyses and how the results from these bring significant added value to current knowledge. The large peptide library was generated 10 years ago and investigated using samples from 28 individuals with latent TB as well as 28 healthy controls. It appears as the current study use a similar approach, but with ATB patients instead. Some of the LTBI data are also included for comparisons. It is unclear when and how the different cohorts were recruited, and if the study was systematically designed to include ATB patients "mid-treatment" and if so, what the benefit of that would be. From this study it is not clear what represents immune correlates of protection or if it practical and relevant to use the ATB116 peptide pool to differentiate ATB from LTBI or healthy controls.

We have now clarified when the different patient cohorts were recruited. The samples used in screening the entire peptide library were recruited from the end of 2012 until the end of 2013. Our approach measures memory T cell responses. To measure reactivity against ~20k peptides, many PBMCs are needed. It is not advisable for individuals recently diagnosed with ATB to undergo leukapheresis. Therefore patients mid-treatment were included since they were well enough to provide leukapheresis samples. As described in the manuscript, the ATB116 peptide pool defined in patients mid-treatment was also reactive in patients who donated blood at the time of their diagnosis. This study was not designed to identify correlates of protection, but the ATB116 peptide pool may be a useful tool (similar to the ESAT-6 and CFP10 used in IGRA assays) to

distinguish between ATB from LTBI and uninfected controls in similar assays as the IGRA tests.

Specific comments:

1. The rationale for using PBMC samples from ATB patients “mid-treatment” needs to be motivated. Already after intensive-phase treatment 2 month after start of treatment, the Mtb load and secreted antigens may be very different compared to ATB patients at diagnosis. This is of particular importance as the longitudinal flow data in Figure 6, suggest that the CD4+ cytokine responses are quite different comparing visit 1, 2 and 3. Then the IFN γ responses in the Sri Lankan cohort included ATB patients recruited at diagnosis and Moldovan ATB patients recruited at diagnosis and mid-treatment.

Mid-treatment samples: Measuring reactivity against ~20k peptides requires a large number of PBMCs per subject tested. Leukapheresis results in a large number of white blood cells while not influencing (to the same extent) the red blood cells as a whole blood donation would. Patients with active TB at diagnosis are not eligible for a leukapheresis donation. This is due to many factors, including the overall poor health commonly associated with active TB infection and the high infectivity of patients during the early phases after diagnosis. Importantly, stimulation with a peptide pool measures memory T cell responses, and as such, we hypothesized that samples from individuals mid-treatment could be used to allow us to screen the entire peptide library in a single participant. The individuals recruited mid-treatment had a reduction in clinical symptoms, responded well to treatment, and regained weight lost due to the infection, making them eligible for a leukapheresis donation. This information has been clarified in the materials and methods section (study participants) and in the results section line 112.

We agree with the reviewer that both the antigenic load and secreted antigens may be different mid-treatment vs. at the diagnosis. This is also why the ATB116 pool was subsequently validated in individuals at diagnosis. Of course, other peptides in the library may be reactive in individuals with ATB at diagnosis, but performing the same type of screen in those samples is impossible.

I’m also curious how the sample sizes were determined.

Sample sizes were based on availability in the different cohorts. We included all the samples that were available.

In the MM section it is explained that 164 Peruvian patients were recruited; so why were only 21 included in the peptide analyses? How were these 21 patients selected among the total of 164? And how would the data presented in Figure 1 change if a higher number of patients would have been included? Overall, please add the time frame and

years in which the different patient cohorts were recruited. A supplementary table to clarify this may also help.

To clarify, the total number of participants recruited from all three sites was 184 (we have included additional participants from the Moldovan cohort in the revised version of the manuscript). The total number of participants from Peru was the 21 patients that were included in the screen.

If additional participants had been available for the screening, we predict that additional peptides would have been identified that are recognized by multiple individuals.

The time frame of recruitment for each cohort is now included in the materials and methods section (study participants).

2. For the ICS in Figures 4 and 5, how were the sub-group of n=10 patients chosen? Why not analyse all 21 ATB patients? The font size in Figure 4 is too small and should be increased. Furthermore, it is not appropriate to use a Mann-Whitney test to compare groups of three (PC85, PC71, MTB300) but a Kruskal-Wallis test should be used. Similarly, pair-wise comparisons using Mann-Whitney in Figure 5 is not correct.

We randomly selected 10 individuals from the cohort for an unbiased analysis. We have changed the statistical test to Kruskal Wallis with Dunn's multiple test.

3. For the sensitivity and specificity calculations in Table 1, ROC curves should be added to illustrate the data. The fact that the Sri Lankan cohort display 62,5% sensitivity and the Moldovan cohort 65-70% specificity should be discussed. How applicable are these results for diagnostic purposes?

We thank the reviewer for their suggestion to add ROC curves (now figure 6).

As pointed out by the reviewer, the sensitivity was lower in the Sri Lankan cohort compared to the Moldovan, while the specificity is similar between the cohorts (slightly lower in Moldova). This may reflect differences in the infecting Mtb strains, environmental differences, and also potentially host factors. Our results suggest a diagnostic potential for the ATB116 pool, but this will need to be validated in larger cohorts, varied patient groups and different geographical locations. This is discussed in the discussion section line 340.

4. As most of these data are build on the IFN γ response to different peptide pools (IL-17 was undetectable?) it would be good if the authors discussed IFN γ , TNF α and IL-2 as representative correlates of protection and any new literature related to these markers. Is it good to have higher proportions of effector memory cells expressing IFN γ and TNF- α ? Are these T cells likely to be protective, and if so how?

Yes, IL-17 was undetectable, which is now more prominently stated in the results section.

Given the specificity of the peptides in ATB116, i.e., exclusively reactive in patients with active TB, we hypothesize that those responses are not protective.

The question of whether IFN γ , IL-2 and TNF α responses are protective is difficult, various studies examining polyfunctional CD4 $^+$ T cells in relation to host containment of Mtb infection have been contradictory. On one hand, stronger polyfunctional CD4 $^+$ T cell responses specific to mycobacteria have been observed in adults with sputum smears negative for acid-fast bacilli (AFB) and latent Mtb infection (LTBI) compared to those with AFB smear-positive tuberculosis (TB). Successful TB treatment, which reduces the bacterial load, is also associated with significant increases in the proportion of polyfunctional CD4 $^+$ T cells. On the other hand, there are studies demonstrating a positive correlation between polyfunctional CD4 $^+$ T cell responses and increased bacillary load, showing stronger responses in adults with TB compared to those with LTBI and healthy household contacts of adults with TB. These contradictory results highlight the limitations of correlative studies in determining whether polyfunctionality of CD4 $^+$ T cells plays a causal role in the immune control of the pathogen or simply reflects the underlying bacterial burden.

5. In Figure S1, please show the viability staining of the CD3 $^+$ T cells. What markers were included in the DUMP; CD19 and CD14 only? The DUMP/CD3 plot looks odd.

The dump channel included CD14, CD16, CD19, CD20, and live dead dye (eF506) to exclude dead cells as well as CD3 $^-$ cells (Supplementary table 2). As shown in the gating strategy, Dump negative cells were gated on CD3 to get live T cells, followed by the CD4 and CD8 gates, as shown in Figure S1.

6. For the flow cytometry data, do the authors have the possibility to perform some type of unsupervised analyses of their data such as UMAP to study the clustering of T cell subsets and the expression of different cytokines? I believe this would strengthen the data significantly.

We thank the reviewer for their suggestion, but this is beyond the scope of this study. Our primary focus was to identify ATB-specific epitopes. The ICS experiments described were performed to characterize the responses in terms of their polyfunctionality and memory phenotype. We are investigating the ATB116-specific T cells in more detail using flow cytometry and scRNAseq to determine the specific immune signature and cell subset that responds.

Reviewer #2 (Remarks to the Author):

The authors describe T-cell responses against 21.000 peptides derived from MTB with the aim to differentiate active tuberculosis (ATB) versus healthy individuals. The screening is performed in PBMCs from cohorts gauging IFN-gamma and IL-17 responses. More detailed analysis is performed using a combination of intracellular cytokine staining (ICS) and cell surface marker analysis, e.g. CD45RA / CCR7 in order to defined T-cell subsets (precursor, memory, terminally differentiated T-cells). The authors suggested that T-cell reactivity directed against distinct antigens, such as ATB116 may be able to differentiate between patients with active TB and others.

Specific comments:

1. the participants are addressed as ‘‘ATB patients’’ - I would suggest to rephrase patients with ATB, i.e. so it reflects that it is an individual with a disease and not that the disease defines the individual.

We appreciate the suggestion by the reviewer and agree that the disease doesn't define the individual. We have made the suggested changes in the manuscript.

2. lane 221: TNF alpha production was increased in contrast to IFN, similarly to an older paper from S Kaufmann in Nature. TNFalpha is the cytokine which can still be produced by exhausted immune cells, even if IL-2 and/or IFN-gamma does not happen: is the TNFalpha production associated with terminally differentiated immune cells or T-cells with exhaustion markers ? Please comment.

Could TNFa a more useful cytokine as compared to gauging other cytokines, in gauging biologically relevant T-cell responses against defined MTB antigens? Pls comment.

We found the majority of the cells that produce TNFa (similarly to IFNg and IL-2) are effector memory T cells (Fig 4). So at least when it comes to their memory phenotype, they are not terminally differentiated. No other exhaustion markers were measured as part of this study, but an in-depth characterization of the cells responding to ATB116 will be investigated in future work.

While the proportion of TNFa-producing T cells are higher than IFNg, there is still IFNg production. Based on the observations in our study we can't draw firm conclusions whether TNFa would be more useful, and further investigation is warranted. However, for Fluorospot assays with PBMCs there can be high background of TNFa, since many cells produce this cytokine, and as such IFNg is more useful in those assays.

3. lane 113 and figure 1: IFN and IL-17 production was tested in the screening. I may have missed it, yet I had to a hard time to spot IFN versus IL-17 responses. If I miss that point, the readers may miss it as well.

3.1. Pls comment on IFN-gamma versus IL-17 production in response to the target antigens.

We agree with the reviewer that this needed to be clarified further. We measured both IFN γ and IL-17 production in response to the peptide library. However, we did not detect any IL-17 response upon stimulation with these peptides. As a result, our analysis primarily focused on IFN-gamma production as the basis for our findings. This is now more prominently stated in the first paragraph of the results section (line 116).

3.2. Would different patterns emerge as different qualities of T-cell responses are being analyzed ?

Yes, of course, if other cytokines were measured, different patterns of reactivity may emerge. That was outside the scope of this investigation.

3.3. Are there distinct groups of reactivity defined by high IFN versus high IL-17 or high IL-17 plus IFN γ production in response to candidate peptides ?

As mentioned above, there was no IL-17 production in response to these peptides in this cohort. Based on that, there are no distinct groups of reactivity defined by IFN γ and IL-17 together.

3.4. Pls provide an introduction why IFN-gamma and IL-17 was chose as the screening

We have selected IFN-gamma and IL-17 as both of these cytokines play role in TB pathogenesis. It was shown in many research articles that IFN γ has an essential role in the protective immunity to mycobacteria. It has also been demonstrated that the BCG vaccine and purified protein derivative (PPD) are able to expand memory CD4+IL-17+ cells. Furthermore, IL-17+ T cells have been described in active TB patients, particularly in those infected by MDR Mtb strains, suggesting a pathogenic role for this cytokine. Therefore, we chose these two cytokines for screening the epitopes. The rationale for this selection is now included in the first paragraph of the results (line 114).

3.5. Were they also commonly shared targets e.g. CMV or EBV included in the screening and would these targets show differences in patients with ATB ? (see comments in point 4 concerning anergy).

This is an interesting question, but we have not included peptide pools specific for CMV and/or EBV in this study.

4. The testing cohort were 24 patients with ATB from Peru, the blood was harvested 3-4 months after diagnosis from Sri Lanka and Moldavia: 24 patients with ATB from each study place yet blood was collected at the time of diagnosis.

We apologize for the lack of clarity in the methods section. From Peru, 21 individuals were recruited, and they donated blood 3-4 months post-diagnosis of their ATB infection. We do not have access to PBMCs from the time of diagnosis in the Peruvian cohort. The individuals in Sri Lanka were recruited later, and they donated blood at diagnosis (and some also at 2 and 6m post-diagnosis). For Moldova, individuals either at diagnosis of their ATB infection or mid-treatment were recruited.

4.1. Was there an association with high IFN-gamma responses and the Peru cohort ? Since T-cells from patients with active TB harvested at the time of diagnosis show often anergy and reduced immune - reactivity against antigens in general. Please show the responses based on the PBMC harvest (3/4 months versus at the time of diagnosis of ATB).

As mentioned above, we do not have access to at diagnosis samples from the Peruvian cohort. The only samples we could measure responses in at diagnosis and mid-treatment were from the individuals recruited in Sri Lanka and Moldova who provided smaller volume donations. As shown in figure 5 and 7, if anything, higher IFN γ responses were observed at diagnosis compared to mid-treatment when stimulated with the ATB116 peptide pool. Therefore, anergy is not evident in these samples. However, it should be noted that the cohort in Moldova is cross-sectional, and as such, we do not have paired samples at diagnosis and mid-treatment.

4.2. Were there differences in cytokine productions in a) PBMCs harvested at the time of diagnosis versus later concerning IFN-gamma and IL-17, and b) in ICS, i.e. examining IL-2, TNFalpha and IFN-gamma ?

(a) The peptide pool screening and epitope identification were performed only in samples recruited mid-treatment. Therefore, we did not measure IFN γ and IL-17 responses in ATB individuals at diagnosis. IL-17 was also not included in the ICS panel. Therefore, we are unable to comment on whether there are temporal differences between IFN γ and IL-17.

(b) IFN γ , IL-2 and TNF α responses were similar when cells were stimulated with the peptide pools corresponding to different protein categories (fig. 4).

We only observed significant differences for IFN γ in ICS when comparing responses between participants with ATB and IGRAneg participants after stimulation with ATB116 (Fig 5b). Finally, while IFN γ decreased from diagnosis to the second blood draw, TNF α increased and IL-2 showed a trend for increase (Fig 7), in response to ATB116.

4.3. I am confident that ATB does along with at least an Xray. were the study participants stratified based on extent of the disease based on the XRays (more extensive disease is usually associated with increased immunosuppression)?

We agree with the reviewer that this is an important consideration to take into account for future investigations to subdivide patient populations further. In our present study, we do not have access to the X-ray data for the participants. The participants in Peru had culture-confirmed pulmonary TB, and individuals from Sri Lanka and Moldova had a positive GenXpert, positive sputum smear, and/or a positive culture.

4.4. Pls provide data whether TB was sensitive to the conventional antibiotic regimens or XDR or MDR TB (or total-drug resistant).

No patients with XDR or MDR-TB were included in the study. This information has been added to the methods section (line 397).

4.5. is there any bias in the cohort based on different exposures and genetic background: pls show Figure 1 in all its elements based on the three different cohorts (Peru, Sri Lanka, Moldova). If all looks the same, pls comment and move it in the supplementary data sets.

Only individuals from Peru were included in the data generated for Figure 1. Therefore, the bias in terms of different exposures and genetic background is very limited. However, the differences between cohorts (exposure, geographical location, genetic background, etc.) is why we included Sri Lanka and Moldova to validate the reactivity against the ATB116 peptide pool in subsequent experiments.

5. lane 438: the patients who donated 100mL blood underwent a leucapheresis ? This would be very unusual for 100 mL.

Individuals from Peru underwent leukapheresis donation. This process involves passing the blood through a separation (leukapheresis) to get a sample with white blood cells. The total volume of this sample was 100mL after the completion of their donation. They did not donate whole blood and a leukapheresis sample at the same time.

6. What is the biological meaning of T-cell reactivity to the target antigen associated with ATB ?

Based on our results, there seems to be a variation in the reactivity of different T cell epitopes based on the disease state in tuberculosis. The metabolic state of Mtb undergoes a shift during infection, transitioning from active replication to slow or non-

replication. This metabolic shift is accompanied by changes in the gene expression profile, leading to alterations in protein expression and the antigens presented to the immune system. As a result, it is plausible to expect that certain antigens exhibit higher expression levels during the active phase of infection. Consequently, these antigens may elicit a stronger immune response in individuals with active tuberculosis (ATB) compared to those with other disease states. It may be beneficial to include these antigens in a vaccine to boost responses against the antigens that are targeted by T cells during active infection. Moreover, it may help in designing diagnostic tests to distinguish different stages of Mtb infection. These considerations are included in the discussion.

7. What is the clinical consequence and what would be the advantage to run an immunological test (as compared an IGRA, Xray, culture or PCR GenExpert) ?

Currently, we are unable to provide definitive insights into the clinical implications or advantages of implementing a new immunological test. However, our efforts to identify TB disease stage-specific antigens through screening a comprehensive proteome-wide library hold promise for future diagnostic test development. This approach has the potential to distinguish between active and latent TB infections, which the IGRA test is currently unable to distinguish between. Additionally, identifying these stage-specific antigens may have implications for vaccine design, as they can serve as potential targets for developing vaccines specifically targeting active TB. While further research is necessary to fully explore these possibilities, our findings offer promising prospects for advancing diagnostic and vaccine development strategies in tuberculosis.

8. PBMCs after thawing: what was the viability of T-cells AT THE TIME THE FUNCTIONAL ASSAY WAS STARTED. This is important since T-cells from patients with a heavy disease burden usually don't freeze very well - and tend to produce more TNFalpha as compared to other cytokines.

We agree with the reviewer that this is an important consideration. Using Trypan blue and a hemocytometer, the viability was determined right before the functional assay. The viability was >60% for all samples. This information has been added to the methods section (line 411).

Reviewer #3 (Remarks to the Author):

*I have carefully read the paper by Panda et al, and have a number of concerns:
- IL-17 is mentioned in the abstract and methods but not in de the actual data, the data should be described or all mentions should be removed from the manuscript*

We agree with the reviewer and have expanded on the IL-17 results in the manuscript. We included IL17 in our peptide library screening, recognizing its significance in TB pathogenesis. However, intriguingly, we did not observe any IL17 response in our ATB cohort during the library screening.

- The data from the IGRA+ participants to the 1036 peptide pools were previously collected, since the publication is already from 2013 the data must be even older. How do the authors guarantee translatability of these data and is the alignment with the ATB patients in the current paper made, in particular with the direct comparison that is made in Figure 2D and the magnitude of the responses.

The reviewer is correct that the results for the IGRA+ participants were published in 2013. In this study, we measured reactivity against the same peptide sequences in a newly synthesized library to make sure that the peptides had not degraded due to long-term storage. Peptide pools are also aliquoted to avoid excessive freeze/thaw cycles. Therefore, we still think it is valid to compare with the IGRA+ data previously generated, to highlight differences. The identified peptide pools with newly synthesized peptides were subsequently tested in other cohorts to validate the differences observed.

- Figure 2D is hard to read and data should be plotted in a different way.

We changed the order of the graphs in Figure 2 to allow 2D to be shared as a larger graph. Hopefully, this helped with the readability.

- The data on the 517 peptides from the TB vaccine candidates seems unrelated to the 20,610 peptides described for the rest of the data, also the ATB116 peptides are derived purely from the 20,610 peptides, so what is the added value of these data? Why are they included here? The selection process of the peptides to be taken forward is also not completely clear and the description should be extended. It is mentioned that the ATB116 are non-overlapping with the MTB300 peptides but is it unclear in what stage this has been selected, was that already done before screening or were screening results further tailored to arrive at the 116 peptides?

We have now clarified the peptide selection process further in the results section (starting on line 220, 226). The ATB116 peptides are derived from the results of both the proteome-wide screen and the TB vaccine candidates. The ATB116 are non-overlapping with MTB300 since they are defined as “ATB-specific epitopes”. MTB300 was constructed based on the reactivity of epitopes in IGRA+ individuals following the previous proteome-wide screen, and therefore, by definition, they do not overlap with ATB116.

There was no pre-selection before the proteome-wide screen. As described, the proteome-wide screen, and the TB vaccine and IGRA antigens resulted in recognition of a total of 174 peptides (Table S1). Some of them overlap with peptides found in MTB300 (now included in Table S1). None of the peptides included in ATB116 have previously been shown to elicit an immune response in a participant without ATB.

- Overall, not all ATB patients, either at mid-treatment or at diagnosis, recognize the ATB116 pool, but approximately 67% does. How do the authors envision this to be applied in diagnostic testing? In particular because also the IGRA+/- individuals from Moldova with a recent exposure as household contact seem to have the highest background responses, and those type of individuals would be the population to be tested in diagnostic work up? Wouldn't that decrease the value of ATB116 for diagnosis if recent exposure can induce similar responses? This should be discussed more extensively.

Yes, it is true that not all patients with ATB recognize the ATB116 pool, but the same is true for the IGRA test. More investigation is needed to explore the ATB116 as a potential diagnostic, but we were encouraged by our results in the Sri Lankan and Moldovan cohorts. The reactivity in the household contacts could be a sign of recent exposure or, alternatively, a sign of the individual developing active TB. This has been added to the discussion (line 319). We are interested in investigating this further to see whether reactivity to ATB116 predicts progression to ATB (as detailed in the discussion, line 343), and at what time point this occurs. Future studies will determine whether ATB116 is useful as a diagnostic. In any case, an ATB-specific peptide pool may be useful for the scientific community measuring immune responses against Mtb in diverse patient cohorts. ATB116 can be combined with MTB300 to capture more of the total response.

- Figure 4 should be described better, it is unclear which donors were used here, which cohort and which disease state?

In figure 4, we selected 10 donors from the same cohort from Peru from which the screening was done. This has now been clarified in the text as well as in the figure legend.

- The cytokine production and memory states were determined towards 2 peptide pools (Figure 4), PC71 and PC85, but it is unclear what the relation between those peptide pools and the ultimate ATB116 pool is. Why is MTB300 used as a comparator here and not ATB116?

The epitopes identified in the Peruvian cohort in the proteome-wide and TB vaccine and IGRA antigen screen were categorized based on their functional category. The majority of these epitopes (n=71) were derived from the cell wall and cell processes. The remaining

epitopes were combined into a single group encompassing other functional categories. This categorization aimed to investigate whether there were any variations in T cell functionality when epitopes were divided based on cell wall and cell processes vs. other functional categories (indicated in Table S1). As shown, the comparison between the two pools revealed that they trigger a similar T cell response. These experiments were performed (as clarified in the previous question) in the same cohort as the proteome-wide screen, before the establishment of ATB116. We therefore used MTB300 as a comparator, to determine the overall reactivity against a previously well-established Mtb peptide pool. ATB116 is a subset of peptides found in both PC71 and PC85.

- Information on ethical permission for the Sri Lankan cohort is missing

This has been clarified in the methods section. The ethical permission for Sri Lanka is mentioned in line number 373.

- The cytokine data are described in the methods to have a cut off of 0.0001%, which is only 1 cell if starting with 1x10e6 PBMCs... please check if this has been truly applied.

We agree with the reviewer that this is an unreasonable cut-off. We have revised our figures to a cut-off of 0.001% which did not alter the results.

Reviewers' Comments:

Reviewer #1:

Remarks to the Author:

The authors have addressed most of my specific comments and concerns. However, I still feel that the overall message and conclusion of the study must be clarified and enhanced. For example, all sub-titles in the Result section are descriptive. I suggest changing these to highlight the most important finding(s) that can guide the reader. In addition, I believe that both the Abstract and conclusions could be rewritten to better highlight the aim and important findings of this study, the text is a bit thin and mainstream now. Eg. in the abstract "Tuberculosis caused by Mycobacterium tuberculosis is a significant global health concern." or "Responses were also measured against TB vaccine candidates and ESAT-6/CFP10 peptides." could be omitted in favour of describing the results on antigen-specific T cell responses and ROC curve analyses and why this is new and important. I would also omit IL-17 in the Abstract, as it only creates confusion and no data on IL-17 was eventually provided.

Overall, focus on the following:

What was the main aim with the study?

What are the main findings of the study and how are the data applicable?

What is the news value of these findings that have not been shown in similar studies using similar approaches? Eg. how can you convince the audience that this is not just another study on an array of different peptides looking at IFN-gamma production by T cells in response to antigens?

Minor comment is that one of the p-values in Fig 4G, is still bolded as significant $p < 0.085$, but should not be bold.

Reviewer #3:

Remarks to the Author:

The authors have addressed my comments and concerns.

This reviewer was also asked to comment on the authors response to the concerns of reviewer 2 who is not able to provide a response during this round of review.

I have assessed the responses to the questions from reviewer 2 in detail.

My assessment of the answers is:

1. Adapted completely throughout manuscript
2. Answered in rebuttal, not possible to include analysis of TNFa in retrospect
- 3.3.1 Answered (same comment as one of my comments)
- 3.2 authors agree, but consider this outside of the scope and I agree that this would be out of scope
- 3.3 there was no IL-17 detected so impossible to perform this analysis
- 3.4 answered and included in revised manuscript
- 3.5 this was a question from the reviewer which is answered in the rebuttal but not done and thus no data included (asked if also CMV or EBV responses were measured, which would have been nice, but this was not done), which I think is acceptable
4. better description included in the manuscript
- 4.1 answered in rebuttal and explained in more detail in methods (confusion on sample collection)
- 4.2 answered but not included in this level of detail in revised manuscript
- 4.3 the authors indicate not to have access to x-ray images, which is surprising, but I think having images would not change anything to the message of this paper and would be more suitable to study separately in other cohorts
- 4.4 answered
- 4.5 answered
5. answered and included in revised manuscript
6. answered and included in revised manuscript

7. answered

8. answered and included in revised manuscript

Thus in my opinion all issues raised by reviewer 2 have been answered, most have been included in the revised manuscript, but for some the data is not available. To my opinion these are not major issues but would have been nice to add.

Point-by-point reply to reviewer's comments.

Reviewer #1 (Remarks to the Author):

The authors have addressed most of my specific comments and concerns. However, I still feel that the overall message and conclusion of the study must be clarified and enhanced. For example, all sub-titles in the Result section are descriptive. I suggest changing these to highlight the most important finding(s) that can guide the reader. In addition, I believe that both the Abstract and conclusions could be rewritten to better highlight the aim and important findings of this study, the text is a bit thin and mainstream now. Eg. in the abstract "Tuberculosis caused by Mycobacterium tuberculosis is a significant global health concern." or "Responses were also measured against TB vaccine candidates and ESAT-6/CFP10 peptides." could be omitted in favour of describing the results on antigen-specific T cell responses and ROC curve analyses and why this is new and important. I would also omit IL-17 in the Abstract, as it only creates confusion and no data on IL-17 was eventually provided.

Overall, focus on the following:

What was the main aim with the study?

What are the main findings of the study and how are the data applicable?

What is the news value of these findings that have not been shown in similar studies using similar approaches? Eg. how can you convince the audience that this is not just another study on an array of different peptides looking at IFN-gamma production by T cells in response to antigens?

We thank the reviewer for their suggestions and have rewritten the abstract, sub-titles in the results section and the conclusions.

Minor comment is that one of the p-values in Fig 4G, is still bolded as significant $p < 0.085$, but should not be bold.

This has been corrected.

Reviewer #3 (Remarks to the Author):

The authors have addressed my comments and concerns.

This reviewer was also asked to comment on the authors response to the concerns of reviewer 2 who is not able to provide a response during this round of review.

We thank the reviewer for also considering our responses to reviewer 2. We have added additional information to the manuscript regarding their question 4.2, as

suggested. Specifically, we now include in the discussion: “No IL-17 was detected in the individuals with ATB mid-treatment, but this does not rule out differences between IFN γ and IL-17 at diagnosis of ATB, where we have not measured IL-17.” And “IFN γ , IL-2, and TNF α responses were similar when cells were stimulated with the peptide pools corresponding to different protein categories. In the longitudinal samples. IFN γ decreased from diagnosis to the second time point 2m post diagnosis, TNF α increased, and IL-2 showed a trend for an increase in response to ATB116.”

I have assessed the responses to the questions from reviewer 2 in detail.

My assessment of the answers is:

- 1. Adapted completely throughout manuscript*
- 2. Answered in rebuttal, not possible to include analysis of TNF α in retrospect*
- 3.3.1 Answered (same comment as one of my comments)*
- 3.2 authors agree, but consider this outside of the scope and I agree that this would be out of scope*
- 3.3 there was no IL-17 detected so impossible to perform this analysis*
- 3.4 answered and included in revised manuscript*
- 3.5 this was a question from the reviewer which is answered in the rebuttal but not done and thus no data included (asked if also CMV or EBV responses were measured, which would have been nice, but this was not done), which I think is acceptable*
- 4. better description included in the manuscript*
- 4.1 answered in rebuttal and explained in more detail in methods (confusion on sample collection)*
- 4.2 answered but not included in this level of detail in revised manuscript*
- 4.3 the authors indicate not to have access to x-ray images, which is surprising, but I think having images would not change anything to the message of this paper and would be more suitable to study separately in other cohorts*
- 4.4 answered*
- 4.5 answered*
- 5. answered and included in revised manuscript*
- 6. answered and included in revised manuscript*
- 7. answered*
- 8. answered and included in revised manuscript*

Thus in my opinion all issues raised by reviewer 2 have been answered, most have been included in the revised manuscript, but for some the data is not available. To my opinion these are not major issues but would have been nice to add.

Reviewers' Comments:

Reviewer #1:

Remarks to the Author:

Thank you, the authors have responded to all my commenst and I belive the manuscript has improved significantly after revision.